# A first-order primal-dual method with adaptivity to local smoothness

**Maria-Luiza Vladarean**[⋆]      **Yura Malitsky**[†]      **Volkan Cevher**[⋆]

{maria-luiza.vladarean, volkan.cevher}@epfl.ch
yurii.malitskyi@liu.se

[⋆] LIONS, École Polytechnique Fédérale de Lausanne, Switzerland
[†] Linköping University, Sweden

## Abstract

We consider the problem of finding a saddle point for the convex-concave objective $\min_x \max_y f(x) + \langle Ax, y \rangle - g^*(y)$, where $f$ is a convex function with locally Lipschitz gradient and $g$ is convex and possibly non-smooth. We propose an adaptive version of the Condat-Vũ algorithm, which alternates between primal gradient steps and dual proximal steps. The method achieves stepsize adaptivity through a simple rule involving $\|A\|$ and the norm of recently computed gradients of $f$. Under standard assumptions, we prove an $\mathcal{O}(k^{-1})$ ergodic convergence rate. Furthermore, when $f$ is also locally strongly convex and $A$ has full row rank we show that our method converges with a linear rate. Numerical experiments are provided for illustrating the practical performance of the algorithm.

## 1 Introduction

In this paper we study a particular instance of the composite minimization problem

$$\min_{x \in \mathcal{X}} f(x) + g(Ax), \tag{1}$$

where $f$ and $g$ are convex, proper and lower-semicontinuous (l.s.c.), and $A$ is a linear operator.

Problems of the form (1) have been studied in the literature under various assumptions on $f$ and $g$. For the particular instances where $g \circ A$ is proximal-friendly and $f$ is $L$-smooth, the objective is suitable for applying forward-backward splitting algorithms like the Proximal Gradient algorithm and its accelerated counterpart [Nesterov, 2013, Beck and Teboulle, 2009]. In general, however, the proximal operator of $g \circ A$ is not easily computable and in such cases a popular approach is to decouple $A$ and $g$ by reformulating problem (1) as a convex-concave saddle-point problem:

$$\min_{x \in \mathcal{X}} \max_{y \in \mathcal{Y}} \langle Ax, y \rangle + f(x) - g^*(y), \tag{2}$$

where $g^*$ denotes the Fenchel conjugate of $g$. Objective (2) is typically addressed by primal-dual splitting algorithms which, under strong duality, can recover the solution to the original problem (1). In the particular case when $f$ and $g$ are proximal-friendly and possibly non-smooth, a very popular method is the Primal-Dual Hybrid Gradient proposed in [Chambolle and Pock, 2011], which was further extended to handle an additional $L$-smooth component with the Condat-Vũ algorithm [Condat, 2013, Vũ, 2013]. Convergence rates for the latter are studied in [Chambolle and Pock, 2016a].

Together, these classes of algorithms cover a broad range of problems in diverse fields such as signal processing, machine learning, inverse problems, telecommunications and many others. As a result, a great amount of research effort has gone into addressing practical concerns such as robustness

to inexact oracles, acceleration and automation of stepsize selection. For a comprehensive list of examples and theoretical details we refer the reader to review papers [Combettes and Pesquet, 2011, Parikh and Boyd, 2014, Komodakis and Pesquet, 2015, Chambolle and Pock, 2016b]. In this work, we focus on the line of investigation studying stepsize regime automation for primal-dual algorithms targeting problem (2).

In their basic form, primal-dual methods require as input stepsize parameters belonging to a designated interval of stability, which depends on problem specific constants like the global smoothness parameter $L$ and $\| A \|$. Dependence on such constants is undesirable because they may be costly to compute and oftentimes one can only access upper-bound estimates, thus leading to overly-conservative stepsizes and slower convergence. Moreover, the need to know $L$ for setting the stepsizes prevents these methods from being applied to functions which are not globally smooth.

Consequently, recent efforts have gone towards devising methods with adaptive stepsizes [Goldstein et al., 2013, 2015, Malitsky and Pock, 2018, Pedregosa and Gidel, 2018]. These approaches resort to linesearch for finding good stepsizes at every iteration, and exhibit improved practical performance. It thus appears that better convergence comes at the cost of an indeterminate number of extra steps spent in subprocedures aimed at finding appropriate stepsizes.

In this work, we study problem (2) under the assumption that $\nabla f$ is locally Lipschitz continuous and $g$ is proximal-friendly. To illustrate the motivation of our framework, we take a prototypical example in image processing:

$$\min_x \frac{1}{2} \| Kx - b \|^2 + \lambda \| Dx \|_{2,1}, \quad K \in \mathbb{R}^{m \times d}, D \in \mathbb{R}^{2d \times d},$$

where $x$ is an image, $K$ is a problem-specific measurement operator, $b$ is the vector of (possibly noisy) observations and $D$ is the discrete gradient operator and the regularization term represents the isotropic TV norm. In order to apply any of the aforementioned primal-dual algorithms, one needs to first choose how to decouple the linear operators. There are three options: decoupling with respect to $K$ leaves us with having to compute the proximal operator of the TV norm for the primal step, which is an iterative procedure [Chambolle, 2004]. Decoupling $D$ implies performing gradient steps on $f$, since in general its proximal operator is not efficient. Finally, decoupling with respect to both implies increasing the dimensionality of the dual variable to $m + 2d$, which is problematic for large $d$ and $m$. The sensible choice is the second one (i.e., decoupling $D$), and the question we seek to answer with this work is:

> *Does there exist a method for solving* (2) *that adapts to the local problem geometry without resorting to linesearch?*

Our contribution is to propose a first-order primal-dual scheme that answers this question in the affirmative and is accompanied by theoretical convergence guarantees. Using standard analysis techniques we show an ergodic convergence of $\mathcal{O}(k^{-1})$ when $\nabla f$ is *locally* Lipschitz and $g$ is proximal-friendly, and a linear convergence rate for the case when $f$ is in addition *locally* strongly convex and $A$ has full row rank. We provide numerical experiments for sparse logistic regression and image inpainting, as well as use our method as a heuristic for TV-regularized nonconvex phase retrieval.

The rest of the paper is structured as follows: Section 2 provides details about related work; Section 3 introduces notation, along with technical preliminaries and assumptions to be used in our analysis; Section 4 reports the main theoretical results alongside partial proofs; finally, partial numerical results are provided in Section 5 with the rest being deferred to the appendix due to lack of space.

## 2  Related Work

**Adaptive Gradient Descent (GD) methods.**   Arguably the most widespread of optimization methods, GD presents similar shortcomings for setting the stepsize as those described in the previous section. In particular, much research effort has gone in devising variants of the algorithm that remove the need to estimate the global smoothness constant $L$. In a recent work, Malitsky and Mishchenko [2020] propose an extremely simple and effective alternative for setting the stepsize $\tau_k$ adaptively at every iteration, as follows:

$$\tau_k = \min \left\{ \tau_{k-1} \sqrt{1 + \frac{\tau_{k-1}}{\tau_{k-2}}}, \frac{\| x_k - x_{k-1} \|}{2 \| \nabla f(x_k) - \nabla f(x_{k-1}) \|} \right\}. \tag{3}$$

Adaptivity essentially comes 'for free' in (3), as it involves solely quantities which have already been computed. Moreover, the method requires only the weaker assumption of local smoothness, thus extending the reach of provably-convergent GD to a wider class of differentiable functions while maintaining the standard $\mathcal{O}(k^{-1})$ convergence rate.

In this work we show that the above technique can be extended to the analysis of primal-dual methods, where it gives rise to an algorithm whose stepsizes adapt to the local geometry of the objective's (locally) smooth component $f$.

**Adaptive monotone variational inequality (VI) methods.** Malitsky [2020] proposes an algorithm for solving monotone VIs with a stepsize that adapts to local smoothness similarly to (3). This method solves the very general formulation of finding $u^*$ such that $\langle F(u^*), u - u^* \rangle + h(u) - h(u^*) \geq 0$, $\forall u$ for a given monotone operator $F$ which is locally Lipschitz continuous. Our template (2) can be recovered from theirs by setting $u = (x, y)$, with

$$F(u) = F(x, y) = \begin{bmatrix} \nabla f(x) + A^T y \\ -Ax \end{bmatrix},$$

and $h(u) = g^*(y)$. The advantages of this approach are the relaxed requirement of local Lipschitz continuity for $F$ and the fact that knowledge of $\|A\|$ is not required. However, since the VI framework is very general and does not take advantage of the problem structure (e.g. the fact that $\langle Ax, y \rangle$ is a bilinear term), the method comes with worse convergence bounds than algorithms specifically designed to solve (2). In addition, the algorithm requires as input an upper bound on the stepsizes, despite them being set in accordance to the estimated local smoothness.

**First order primal dual algorithms and adaptive versions.** A popular method for solving (2) when $f$ is $L$-smooth is the Condat-Vũ algorithm (CVA) [Condat, 2013, Vũ, 2013]. The method's convergence is subject to a global stepsize validity condition given by $\left(\frac{1}{\tau} - L\right)\frac{1}{\sigma} \geq \|A\|^2$, where $\tau$ and $\sigma$ are the primal and dual stepsizes, respectively.

Another approach to solving problem (2) is via the Primal–Dual Fixed-Point algorithm based on the Proximity Operator (PDFP$^2$O) or the Proximal Alternating Predictor–Corrector (PAPC) methods [Loris and Verhoeven, 2011, Chen et al., 2013, Drori et al., 2015]. This approach comes with less restrictive stepsize conditions than CVA owing to a different iteration style, but which nevertheless depend on the global smoothness constant $L$ and $\|A\|$ and have to be carefully chosen.

In order to alleviate the burden of choosing the stepsize parameters in CVA, Malitsky and Pock [2018] propose a linesearch procedure involving only dual variable updates and which, for certain problems such as regularized least squares, does not require any additional matrix-vector multiplications. A characteristic of this algorithm is that it maintains a constant ratio between primal and dual stepsizes through a hyperparameter $\beta$ — a setup which we also use in this work.

## 3   Preliminaries

Consider problem (2) and let $\mathcal{X}, \mathcal{Y}$ be finite dimensional real vector spaces equipped with the standard inner product $\langle \cdot, \cdot \rangle$ and the associated Euclidean norm $\|\cdot\| = \sqrt{\langle \cdot, \cdot \rangle}$. We denote by $g^*$ the Fenchel conjugate of $g$ in (1) defined as $g^*(y) = \sup_x \{\langle x, y \rangle - g(x)\}$. In order to not overload the $*$ notation, we use $A^T$ to denote the adjoint operator of $A$.

One can easily see that (2) is a primal-dual formulation of the following primal and dual optimization problems, of which the former is the same as (1):

$$\min_{x \in \mathcal{X}} f(x) + g(Ax), \qquad \max_{y \in \mathcal{Y}} -(f^*(-A^T y) + g^*(y)).$$

A saddle-point $(x^*, y^*) \in \mathcal{X} \times \mathcal{Y}$ of problem (2) satisfies the following optimality conditions:

$$-A^T y^* = \nabla f(x^*), \qquad Ax^* \in \partial g^*(y^*). \tag{4}$$

For $(x', y') \in \mathcal{X} \times \mathcal{Y}$ we define the following quantities:

$$P_{x', y'}(x) := f(x) - f(x') + \langle x - x', A^T y' \rangle,$$

$$D_{x',y'}(y) := g^*(y) - g^*(y') - \langle Ax', y - y' \rangle,$$
$$\mathcal{G}_{x',y'}(x,y) := P_{x',y'}(x) + D_{x',y'}(y).$$

These functions are convex for fixed $(x', y')$ and whenever $(x', y') = (x^*, y^*)$, it holds that $P_{x^*,y^*}(x) \geq 0$, $D_{x^*,y^*}(x) \geq 0$ and $\mathcal{G}_{x^*,y^*}(x,y) \geq 0$, with the latter quantity representing the primal-dual gap. We also define the gap restricted to a bounded subset $B_1 \times B_2 \subset \mathcal{X} \times \mathcal{Y}$ as:

$$\mathcal{G}_{B_1 \times B_2}(x,y) := \sup_{(x',y') \in B_1 \times B_2} P_{x',y'}(x) + D_{x',y'}(y),$$

and note that it is non-negative whenever $B_1 \times B_2$ contains a saddle-point.

Given a function $f : \mathcal{X} \to \mathbb{R}$ and $L > 0$, we say that $f$ is $L$-smooth if its gradient $\nabla f$ is Lipschitz continuous: $\|\nabla f(x) - \nabla f(y)\| \leq L\|x - y\|, \forall x, y$. Furthermore, $f$ is locally smooth if $\nabla f$ is Lipschitz continuous on any compact subset $\mathcal{C}$: $\forall \mathcal{C} \subset \mathcal{X}, \exists L_{\mathcal{C}} > 0$ such that $\|\nabla f(x) - \nabla f(y)\| \leq L_{\mathcal{C}}\|x - y\|, \forall x, y \in \mathcal{C}$.

We also say that $f$ is $\mu$-strongly convex if $f(y) \geq f(x) + \langle \nabla f(x), y - x \rangle + \frac{\mu}{2} \| x - y \|^2, \forall x, y$. Similarly, $f$ is locally strongly convex if it is strongly convex on any compact subset $\mathcal{C}$: $\forall \mathcal{C} \subset \mathcal{X}, \exists \mu_{\mathcal{C}} > 0$ such that $f(y) \geq f(x) + \langle \nabla f(x), y - x \rangle + \frac{\mu_{\mathcal{C}}}{2} \| x - y \|^2, \forall x, y \in \mathcal{C}$.

We define the proximal operator of a convex function $g : \mathcal{X} \to \mathbb{R} \cup \{\infty\}$ as $\mathrm{prox}_g(x) = \arg\min_z \left\{ g(z) + \frac{1}{2} \| x - z \|^2 \right\}$, and say that $g$ is 'proximal-friendly' if $\mathrm{prox}_g(x)$ has a closed form solution or can be efficiently computed to high accuracy.

Finally, the following two blanket assumptions will hold throughout the paper:

**Assumption 3.1.** *Function $f$ is convex and locally smooth, while $g$ convex, l.s.c., and proximal-friendly.*

**Assumption 3.2.** *A saddle-point exists for problem* (2) *and thus strong duality holds.*

We note that Assumption 3.2 is standard in the literature (see e.g., [Chambolle and Pock, 2011]). Assumption 3.1, on the other hand, is weaker than the usual global $L$-smoothness premise and thus enlarges the category of admissible functions $f$ with instances such as $x \mapsto \exp(x)$. To illustrate, consider the aforementioned function defined on the reals: the global smoothness assumption clearly does not hold, however for any fixed interval $[a, b] \subset \mathbb{R}$ the smoothness constant can be chosen as $\exp(b)$.

For showing linear convergence of our method, we will add the following assumption:

**Assumption 3.3.** *Function $f$ is locally strongly convex and operator $A$ has full row-rank.*

## 4   Algorithm and convergence

The primal-dual method proposed for solving problem (2) under assumptions 3.1 and 3.2, is provided in Algorithm 1 under the abbreviation APDA, which we use from here onwards. APDA follows the same structure as the basic CVA [Chambolle and Pock, 2016a] for the given assumptions. Notice that if we restrict Assumption 3.1 to $L$-smooth functions $f$, we can in fact recover CVA by setting $\theta_k = \theta = 1$ and $\tau_k = \tau, \sigma_k = \sigma$ fixed such that $\left(\frac{1}{\tau} - L\right) \frac{1}{\sigma} \geq \| A \|^2$.

---

**Algorithm 1** Adaptive Primal Dual Algorithm (APDA)

> **Input:** $x_0 \in \mathcal{X}, y_0 \in \mathcal{Y}, \tau_{\mathrm{init}} > 0, \tau_0 = \infty, \theta_0 = 1, \beta > 0, c \in (0,1)$
> $x_1 = x_0 - \tau_{\mathrm{init}}(\nabla f(x_0) + A^T y_0)$
> **for** $k = 1, 2, \ldots$ **do**
>    Set $\tau_k = \min \left\{ \frac{1}{2\sqrt{L_k^2 + (\beta/(1-c))\| A \|^2}}, \tau_{k-1}\sqrt{1 + \theta_{k-1}} \right\}, \sigma_k = \beta\tau_k, \theta_k = \frac{\tau_k}{\tau_{k-1}}$
>    $\tilde{x}_k = x_k + \theta_k(x_k - x_{k-1})$
>    $y_{k+1} = \mathrm{prox}_{\sigma_k g^*}(y_k + \sigma_k A\tilde{x}_k)$
>    $x_{k+1} = x_k - \tau_k(\nabla f(x_k) + A^T y_{k+1})$
> **end for**

---

## 4.1 High level ideas

We can rephrase the global stepsize condition of CVA by introducing a free parameter $\beta > 0$ which represents the ratio between the fixed dual and primal stepsizes: $\beta = \frac{\sigma}{\tau}$. With this change of variables, the stepsize validity condition becomes $\tau \in \left( 0, \frac{2}{L + \sqrt{L^2 + 4\beta \| A \|^2}} \right)$.

Our algorithm disposes of CVA's global condition and relies instead on a very similar but *local* criterion given by $\tau_k \in \left( 0, \frac{1}{L_k + \sqrt{L_k^2 + 2\beta \| A \|^2}} \right)$, where $L_k := \frac{\| \nabla f(x_k) - \nabla f(x_{k-1}) \|}{\| x_k - x_{k-1} \|}$ provides an estimate of the local smoothness constant and $\beta = \frac{\sigma_k}{\tau_k}$. In particular, this requirement is satisfied by the first part of the expression defining $\tau_k$ in APDA:

$$\tau_k = \min \left\{ \frac{1}{2\sqrt{L_k^2 + (\beta/(1-c)) \| A \|^2}}, \tau_{k-1} \sqrt{1 + \theta_{k-1}} \right\} \tag{5}$$

where $c \in (0, 1)$. Intuitively, this rule demands that $\tau_k$ does not overstep a constant related to the local curvature, thus allowing for larger stepsizes in flatter regions and correspondingly smaller ones otherwise.

By itself, the first term of (5) does not ensure convergence, since overly-aggressive and possibly destabilizing stepsizes might occur in near-linear regions. This issue is addressed by the second part of the expression (5) which, informally, prevents the stepsize from increasing 'too fast' in consecutive iterations. Specifically, the increase factor is at most $\sqrt{1 + \theta_{k-1}}$, where $\theta_k = \frac{\tau_{k-1}}{\tau_{k-2}}$.

Under these two local stepsize conditions we are able to show APDA's convergence using the weaker assumption of local smoothness of $f$, thus conveniently removing the need of estimating a global smoothness constant $L$.

**Remark 4.1.** *While $\tau_k$ does not adapt to $\| A \|$, for many practical problems this fact is not a big hindrance. Function $f$ typically represents the data fidelity term, whose smoothness constant $L$ (should it exist) can far exceed $\| A \|$ – the linear operator enforcing structured regularization on $x$. A specific example are TV-regularized imaging problems, where $A$ is the discrete gradient operator whose norm is bounded by $\sqrt{8}$ [Chambolle, 2004], while the data fidelity term may involve a very large number of measurements and a larger norm, consequently.*

**Remark 4.2.** *APDA takes an additional primal step prior to the for-loop, which is controlled by $\tau_{init}$ given as input. This is needed for estimating $L_1$ in the first iteration. In practice we set $\tau_{init} = 1e\text{-}9$, a sufficiently small value to ensure that $x_1$ does not depart too far from $x_0$ and yield a good estimate of $L_1$. Furthermore, the setting of $\tau_0 = \infty$ simply ensures that in the first step, $\tau_1 = \frac{1}{2\sqrt{L_1^2 + (\beta/(1-c)) \| A \|^2}}$ and has no impact on further steps. Finally, in our experiments we set $c = 1e\text{-}15$ – this is a parameter introduced for theoretical purposes as explained in the following section.*

## 4.2 Analysis – the base case

In short, the main steps of our analysis are: first, we establish the inequality that characterizes the dynamics of APDA given in Lemma 4.1 below. Based on it, we are able to prove the boundedness of sequences $\{x_k\}$ and $\{y_k\}$ in Theorem 4.1. In turn, sequence boundedness alongside the local smoothness property of $f$ allows us to conclude that there exists a constant $L > 0$ such that $f$ is $L$-smooth on the compact set $\overline{\text{Conv}}(\{x^*, x_0, x_1, \ldots\})$ – the closed convex hull generated by $\{x^*, x_0, x_1, \ldots\}$. Finally, we leverage this information to show that $(x_k, y_k)$ converges to a saddle point of (2) and derive the associated ergodic convergence rates presented in Theorem 4.1.

**Lemma 4.1.** *Consider APDA along with Assumptions 3.1 and 3.2 and $(x, y) \in \mathcal{X} \times \mathcal{Y}$. Then, for all $k$ and $\eta_k \in \left( \frac{\beta \tau_k \| A \|}{1 - c}, \frac{1 - 2\tau_k L_k}{2\tau_k \| A \|} \right)$,*

$$\| x_{k+1} - x \|^2 + \frac{1}{\beta} \| y_{k+1} - y \|^2 + (1 - \eta_k \tau_k \| A \| - \tau_k L_k) \| x_{k+1} - x_k \|^2$$

$$+ \frac{\eta_k - \tau_k \beta \| A \|}{\beta \eta_k} \| y_{k+1} - y_k \|^2 + 2\tau_k (1 + \theta_k) P_{x,y}(x_k) + 2\tau_k D_{x,y}(y_{k+1})$$

$$\leq \| x_k - x \|^2 + \frac{1}{\beta} \| y_k - y \|^2 + \tau_k L_k \| x_k - x_{k-1} \|^2 + 2\tau_k \theta_k P_{x,y}(x_{k-1}).$$

*Moreover, it holds that:*

1) $\tau_k L_k < \frac{1}{2} < 1 - \eta_k \tau_k \| A \| - \tau_k L_k,$

2) $\frac{1}{\beta} - \frac{\tau_k \| A \|}{\eta_k} > \frac{c}{\beta} > 0.$

*Proof sketch.* The full proof is deferred to the appendix. We use algebraic manipulations, APDA's update rules, the Cauchy-Schwarz and Young inequalities and properties of the $\mathrm{prox}$ operator to get the recurrence:

$$\| x_{k+1} - x \|^2 + \frac{1}{\beta} \| y_{k+1} - y \|^2 + (1 - \tau_k \| A \| \eta_k - \tau_k L_k) \| x_{k+1} - x_k \|^2$$

$$+ \left( \frac{1}{\beta} - \frac{\tau_k \| A \|}{\eta_k} \right) \| y_{k+1} - y_k \|^2 + 2\tau_k(1 + \theta_k)P_{x,y}(x_k) + 2\tau_k D_{x,y}(y_{k+1})$$

$$\leq \| x_k - x \|^2 + \frac{1}{\beta} \| y_k - y \|^2 + \tau_k L_k \| x_k - x_{k-1} \|^2 + 2\tau_k \theta_k P_{x,y}(x_{k-1}), \quad (6)$$

where $\eta_k > 0$ is a free iteration-dependent constant involved in Young's inequality.

In order to obtain anything worthwhile we would like to set $\eta_k$ such that, when unrolling (6) over the iterations, the terms containing $\| x_{k+1} - x_k \|^2$ and $\| y_{k+1} - y_k \|^2$ accumulate on the LHS with positive coefficients. More precisely, we ask that:

$$\begin{cases} \frac{1}{\beta} - \frac{\tau_k \| A \|}{\eta_k} > \frac{c}{\beta}, \\ 1 - \tau_k \| A \| \eta_k - \tau_k L_k > \frac{1}{2}, \end{cases} \quad (7)$$

where $c \in (0, 1)$. We note that the RHS of the first inequality could have been chosen as $0$, however, we made it strictly positive due to technical reasons related to controlling the sequence $\| y_{k+1} - y_k \|^2$. In practice, we choose $c$ to be as small as possible.

A similar remark holds for the second inequality, where it would have been sufficient to set its RHS to $\tau_{k+1} L_{k+1}$. Since this would considerably complicate the analysis, we make the observation that $\tau_k L_k < \frac{1}{2}, \forall k$ and use this simpler uniform upper-bound instead.

The inequalities (7) are equivalent to asking that $\eta_k \in \left( \frac{\tau_k \beta \| A \|}{1-c}, \frac{1 - 2\tau_k L_k}{2\tau_k \| A \|} \right)$ and what is left to show is that this is a valid interval i.e., that the left endpoint is strictly smaller than its right counterpart. This condition amounts to solving a quadratic inequality in $\tau_k$, whose solutions lie in the interval $\left( 0, \frac{1}{L_k + \sqrt{L_k^2 + 2(\beta/(1-c))\| A \|^2}} \right)$. The proof is concluded by showing that our choice of $\tau_k$ indeed satisfies this constraint. $\qquad \square$

We are now ready to state the main convergence result in Theorem 4.1 below, whose full proof is given in the appendix.

**Theorem 4.1.** *Consider APDA along with Assumptions 3.1 and 3.2, and let $(x^*, y^*) \in \mathcal{X} \times \mathcal{Y}$ be a saddle point of problem (2). Then, for all $k$*

1) **Boundedness.** *The sequence $\{(x_k, y_k)\}$ is bounded. Specifically, for all k,*

$$\| x_k - x^* \|^2 + \| y_k - y^* \|^2 \leq M,$$

*where $M := \| x_1 - x^* \|^2 + \frac{1}{\beta} \| y_1 - y^* \|^2 + \frac{1}{2} \| x_1 - x_0 \|^2 < \infty.$*

2) **Convergence to a saddle point.** *The sequence $\{(x_k, y_k)\}$ converges to a saddle point of (2).*

*3) Ergodic convergence.* Let $S_k := \sum_{i=1}^{k} \tau_i$, $X_k := \frac{1}{S_k}\left(\tau_k(1+\theta_k)x_k + \sum_{i=1}^{k-1}\left(\tau_i(1+\theta_i)-\tau_{i+1}\theta_{i+1}\right)x_i\right)$ *and* $Y_k := \frac{1}{S_k}\sum_{i=1}^{k}\tau_i y_{i+1}$. *Then, for any bounded* $B_1 \times B_2 \in \mathcal{X} \times \mathcal{Y}$ *and for all* $k$,

$$\mathcal{G}_{B_1 \times B_2}(X_k, Y_k) \leq \frac{M(B_1, B_2)\sqrt{L^2 + (\beta/(1-c))\|A\|^2}}{k},$$

*where $L$ is the Lipschitz constant of $\nabla f$ over the compact set $\overline{Conv}(\{x^*, x_0, x_1, \dots\})$ and* $M(B_1, B_2) = \sup_{(x,y)\in B_1 \times B_2} \|x_1 - x\|^2 + \frac{1}{\beta}\|y_1 - x\|^2 + \frac{1}{2}\|x_1 - x_0\|^2$.

The boundedness result of Theorem 4.1 point 1) implies that the closed set $\mathcal{C} = \overline{Conv}(\{x^*, x_0, x_1, \dots\})$ is also bounded and hence compact. The local smoothness assumption on $f$ then ensures that there exists $L > 0$ such that $f$ is $L$-smooth over $\mathcal{C}$. Note that such an $L$ exists for any $x_0$, $y_0$ since the boundedness result itself holds for any initial conditions (though the value of such $L$ cannot be generally known, as it is path-dependent). Using this fact, we can show a uniform lower-bound on the primal stepsize: $\tau_k \geq \frac{1}{2}\left(L^2 + (\beta/(1-c))\|A\|^2\right)^{-1/2} > 0$, $\forall k$, which is instrumental in deriving the subsequent convergence results, as well as Theorem 4.2. We emphasize that the appearance of constant $L$ in the provided rates is a consequence of iterate boundedness, whose proof does not require its knowledge. Finally, we note that our rate is comparable to that of CVA in terms of constants.

## 4.3 Analysis under the additional Assumption 3.3

We now study APDA under the additional assumption of locally strongly convex $f$ and full row rank $A$. Before proving the result of Theorem 4.2, a few remarks are in order. First, the boundedness result of Theorem 4.1 point 1) also holds for constant $c = 0$, since this constant was required only for proving convergence to a saddle point in point 2) of the theorem. Second, taking a smaller stepsize than the originally defined $\tau_k$ will not change the validity of Lemma 4.1 or the boundedness result of Theorem 4.1, as it remains within the required interval mentioned in section 4.1.

Consequently, for studying APDA under the additional Assumption 3.3 we can simplify the stepsize expression by taking $c = 0$, because now we are able to show iterate convergence directly by using the strong convexity and full row-rank assumptions. Specifically, we consider the stepsize:

$$\tau_k = \min\left\{\frac{1}{2\sqrt{4L_k^2 + \beta\|A\|^2}}, \tau_{k-1}\sqrt{1+\theta_{k-1}/2}\right\}, \tag{8}$$

which is smaller than the one originally considered and, due to the aforementioned remarks it ensures that APDA produces a bounded sequence. It follows that, under the local smoothness and local strong convexity assumptions, there exist constants $L$ and $\mu$ such that $f$ is $L$-smooth and $\mu$-strongly convex over $\overline{Conv}(\{x^*, x_0, x_1, \dots\})$.

The existence of these constants along with $A$ being full row rank, in turn, allows us to derive a strengthened version of the inequality in Lemma 4.1 for $(x, y) = (x^*, y^*)$:

$$\|x_{k+1} - x^*\|^2 + \left(\frac{1}{\beta} + q_1\right)\|y_{k+1} - y^*\|^2 + \left(\frac{1}{2} + q_2\right)\|x_k - x_{k+1}\|^2 + q_3\|y_{k+1} - y_k\|^2$$
$$+ 2\tau_k(1+\theta_k)P_{x^*,y^*}(x_k) + 2\tau_k D_{x^*,y^*}(y_{k+1})$$

$$\leq (1-q_4)\|x_k - x^*\|^2 + \frac{1}{\beta}\|y_k - y^*\|^2 + \left(\frac{1}{2} - q_5\right)\|x_k - x_{k-1}\|^2 + 2\tau_k\theta_k P_{x^*,y^*}(x_{k-1}),$$

where $q_1, q_2, q_3, q_4, q_5 > 0$ are constants given in the appendix. This new inequality represents in fact a contraction, which guarantees the linear convergence rate stated in Theorem 4.2.

**Theorem 4.2.** *Consider APDA along with Assumptions 3.1, 3.2 and 3.3. Let $(x^*, y^*) \in \mathcal{X} \times \mathcal{Y}$ be a saddle point of problem (2). Furthermore, let $\tau_k$ be defined by (8) and let $s := \sqrt{4L^2 + \beta \| A \|^2}$ and $t := \sqrt{4\mu^2 + \beta \| A \|^2}$, where $\mu, L$ are the strong convexity and smoothness constants of $f$ over the compact set $\overline{Conv}(\{x^*, x_0, x_1, \ldots\})$.*

*Then, for all $k$:*

$$\| x_k - x^* \|^2 + \frac{1}{\beta} \| y_k - y^* \|^2 \leq (1 - \min\{p, q, r\})^k M,$$

*where the rate constants are given by:*

$$p = \frac{1}{2}, \quad q = \frac{\mu}{4s}, \quad r = \frac{\beta \sigma_{\min}^2(A)\mu}{\beta \sigma_{\min}^2(A)\mu + 8s^2 t + 4L^2 s},$$

*and* $M = \| x_2 - x^* \|^2 + \left(\frac{1}{\beta} + T\right) \| y_2 - y^* \|^2 + \frac{1}{2} \| x_2 - x_1 \|^2 + 2\tau_1 P_{x^*, y^*}(x_1), \quad T = \frac{\sigma_{\min}^2(A)\mu}{8s^2 t + 4L^2 s}$, *with $\sigma_{\min}(A)$ representing the smallest singular value of $A$.*

A few remarks are in order: first, as a sanity check, we observe that when $A = 0$ we recover the contraction factor of [Malitsky and Mishchenko, 2020] which is equal to $q$.

Second, we make some notes on how our rate compares with existing ones. To our knowledge, there are no explicit results regarding the linear convergence of CVA under assumptions similar to ours (linear rates are usually shown for the 3-component objective without assumptions on $A$ — see e.g., [Chambolle and Pock, 2016a]). However, in the case of $L$-smooth and $\mu$-strongly-convex $f$ and full row-rank $A$, Chen et al. [2013] show the linear convergence of PDFP$^2$O with rate:

$$\|x_k - x^*\|^2 \leq \left(\|x_1 - x_0\|^2 + \frac{1}{\sigma_{\max(A)}}\|y_1 - y_0\|^2\right)\left(1 - \min\left\{\frac{\sigma_{\min(A)}^2}{\sigma_{\max(A)}^2}, \frac{\mu}{L}\right\}\right)^{k-1},$$

The rate presented in Theorem 4.2 has a comparatively worse contraction factor. The reason is that our iteration is set up in the style of CVA, where we essentially have a single stepsize to compute using the rephrasing from Section 4.1. Therefore, $\tau_k$ needs to obey the problem structure with respect to both $L$ and $\|A\|$, resulting in the 'mixed' term appearing in the denominator.

Keeping the above in mind, the interested reader may find in the appendix that constants $q$ and $r$ come from a product between $\tau_k$ and other condition number-related quantities, which is tightly liked to the structure of the main inequality used in the paper. This makes the nice separation of condition numbers achieved in PDFP$^2$O's rate not possible in our case and, it seems, the analysis necessary to achieve this kind of adaptivity comes at the cost of worse constants (the same remark holds for [Malitsky and Mishchenko, 2020]).

PDFP$^2$O, on the other hand, achieves a clean bound by having a different iteration style than CVA, as well as a fundamentally different kind of analysis where the iteration is expressed in fixed-point form to show convergence. In this context the stability conditions on the stepsizes are also relaxed — specifically, $0 < \lambda \leq 1/\sigma_{\max}^2(A)$ and $0 < \gamma < 2L$ in [Chen et al., 2013]. A drawback of this approach, however, is that the algorithm has no rate guarantees when $f$ is only smooth and not strongly convex and only asymptotic convergence is shown. Also, PDFP$^2$O requires 3 matrix-vector multiplications per iteration whereas we only require 2.

## 5 Experiments

We now present some numerical experiments conducted for APDA[1]. Additional problems and results are included the appendix. The experiments were implemented in Python 3.9 and executed on a MacBook Pro with 32 GB RAM and a 2,9 GHz 6-Core Intel Core i9 processor.

---

[1]See https://github.com/mvladarean/adaptive_pda.

The baseline we compare against in this section as well as the appendix is CVA, for which we use Algorithm 1 in [Chambolle and Pock, 2016a] (using $g \equiv 0$). In the particular case of sparse logistic regression we also compare against FISTA [Beck and Teboulle, 2009]. For obtaining $x^*$ we ran one of the algorithms for a large number of iterations.

## 5.1 Sparse binary logistic regression

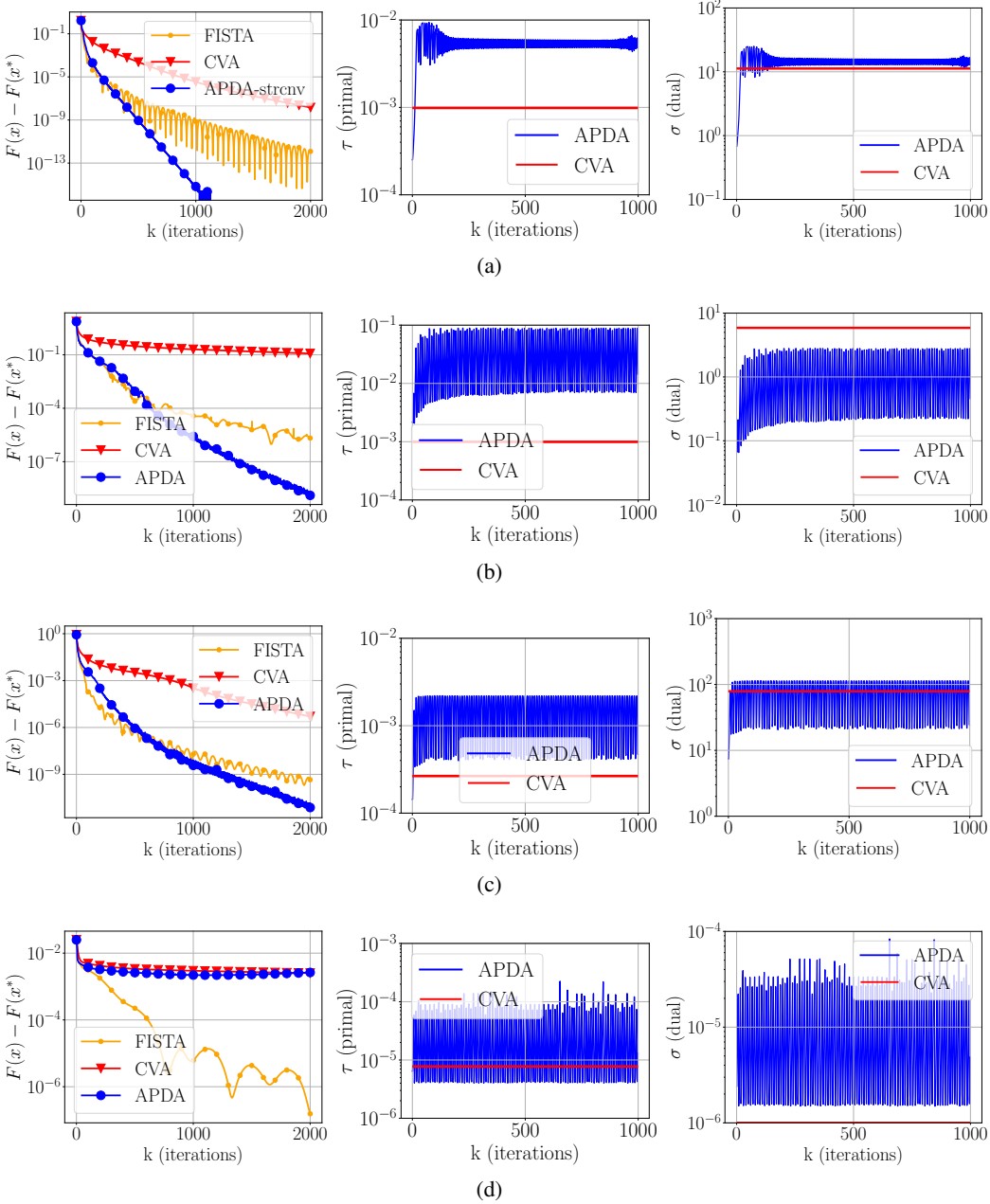

Figure 1: The first column shows algorithm convergence. The second column shows a comparison of primal stepsizes between APDA and CVA. The third column shows a comparison of dual stepsizes between APDA and CVA. Each subfigure represents a different dataset: (a) `ijcnn`; (b) `mushrooms`; (c) `a9a`; (d)`covtype`.

We consider the problem of sparse binary Logistic Regression on 4 LIBSVM datasets [Chang and Lin, 2011] and show that adaptivity provides faster convergence in 3 of these cases. The objective we

consider is:

$$\min_{x \in \mathbb{R}^d} F(x) := \underbrace{\sum_{i=1}^{m} \log(1 + \exp(-b_i \langle q_i, x \rangle))}_{f} + \underbrace{\lambda \, \| x \|_1}_{g}, \tag{9}$$

where $(q_i, b_i) \in \mathbb{R}^d \times \{-1, 1\}$ and $\lambda$ is the regularization parameter. APDA and CVA can be applied to this problem by setting $A = I$ in formulation (2). Primal-dual algorithms are not the typical choice for solving (9), which is usually addressed by methods such as Proximal Gradient or FISTA [Beck and Teboulle, 2009]. However, we note that the computational costs of APDA and FISTA are comparable since the matrix-vector multiplication cost of the former is removed due to a $A = I$.

We choose $\lambda = 0.005 \, \| Q^T b \|_\infty$, where $Q^T = \begin{bmatrix} q_1^T, \ldots q_m^T \end{bmatrix}^T$. For APDA we perform a parameter sweep over $\beta \in [\text{1e-3}, \text{1e6}]$ for each dataset and settle for: $\beta = \text{2.68e3}$ for `ijcnn`; $\beta = \text{5.18e4}$ for `a9a`; $\beta = \text{3.16e1}$ for `mushrooms`; $\beta = \text{3.73e-1}$ for `covtype`.

For CVA we sweep $p \in [\text{1e-3}, \text{1e6}]$ and set $\tau = \frac{1}{\| A \|/p + L}$ and $\sigma = \frac{1}{p \| A \|}$ — by construction, these stepsizes satisfy the validity condition and are as large as possible since the condition is satisfied with equality. We do an additional tuning procedure where we choose constants $\tau \in [\text{1e-10}, \text{1e2}]$ and $\xi \in [\text{1e-5}, \text{1e2}]$ and set $\sigma = \tau \xi$, which are subject to verifying the stepsize validity condition of CVA. Finally we select the best stepsizes across the two tuning phases to be (truncated to 3 decimals): $\tau = \text{9.869e-4}, \sigma = \text{1.125e1}$ for `ijcnn`; $\tau = \text{2.655e-4}, \sigma = \text{7.896e1}$ for `a9a`; $\tau = \text{9.936e-4}$, $\sigma = \text{5.878e0}$ for `mushrooms`; $\tau = \text{7.728e-06}, \sigma = \text{1e-06}$ for `covtype`.

Note that the Hessian of $f$ is given by $\nabla^2 f(x) = Q^T D(x) Q$, where $D(x)$ is a diagonal matrix such that $D_{i,i}(x) = \sigma_i(x)(1 - \sigma_i(x))$, where $\sigma_i(x) = \frac{1}{1 + \exp(-b_i \langle q_i, x \rangle)} \in (0, 1)$. Clearly, over any compact set in $\mathcal{C} \subset \mathcal{X}$ there exist $D_{\min} := \min_{i, x \in \mathcal{C}} D_{i,i}(x) \in (0, 1)$ such that $D_{\min} Q^T Q \preceq Q^T D(x) Q$. As a result, a sufficient condition for local strong convexity is that the minimum eigenvalue of $Q^T Q$ be greater than 0.

The convergence results along with stepsize comparison plots are presented in Figure 1. For dataset `ijcnn` we run APDA with the modified $\tau_k$ used in Theorem 4.2, since $\lambda_{\min}(Q^T Q) = 75.13$ and $A$ has full rank. In the latter case, the legend identifier is `APDA-strcnv`. For the remaining datasets we use only the basic setting for $\tau_k$, as $\lambda_{\min}(Q^T Q) \leq \text{1e-13}$.

While APDA outperforms FISTA and CVA on `ijcnn`, `a9a` and `mushrooms`, it shows a relatively poor performance on `covtype`. We hypothesize that this is related to the condition number of $Q^T Q$, which is almost three orders of magnitude larger in the latter case: `9.2e22` versus `5.3e1`, `2e20` and `2e17` for `ijcnn`, `mushrooms` and `a9a`, respectively. A similar behavior is seen in Figure 1.(c) of [Malitsky and Mishchenko, 2020].

Finally, the adaptive property of APDA's stepsizes is visible in the stepsize comparison plots where they are shown to oscillate within at least one order of magnitude throughout the optimization process.

## 6 Limitations of APDA

The experiments presented in this paper (Section 5 and Appendix A) have the common trait of not imposing hard constraints on the primal variables. As a consequence, we are able to take plain gradient steps in the primal domain. However, for instances such as Poisson linear inverse problems [Bertero et al., 2009], the iterates $x_k$ need to reside in $\mathbb{R}_+^n$ because the primal objective contains $\log$ functions. APDA cannot handle such cases, as any constraints imposed on the primal variables will only be satisfied asymptotically. We consider such scenarios as a future research direction.

## Acknowledgements

The first author is grateful to Ya-Ping Hsieh for his feedback on the manuscript and for helpful research discussions throughout the development of this work. The authors also sincerely thank the anonymous reviewers for their time and their thoughtful, constructive feedback which helped improve and clarify this manuscript.

This project has received funding from the European Research Council (ERC) under the European Union's Horizon 2020 research and innovation programme (grant agreement no. 725594 - time-data), the Wallenberg Al, Autonomous Systems and Software Program (WASP) funded by the Knut and Alice Wallenberg Foundation, with the project number 305286. The work was also sponsored by the Department of the Navy, Office of Naval Research (ONR) under a grant number N62909-17-1-2111; by the Army Research Office and was accomplished under Grant Number W911NF-19-1-0404; by the Hasler Foundation Program: Cyber Human Systems (project number 16066). This work was also supported by the Swiss National Science Foundation (SNSF) under grant number 200021_178865/1.

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
