= \texttt{2.68e3}$ for $\texttt{ijcnn}$; $\beta = \texttt{5.18e4}$ for $\texttt{a9a}$; $\beta = \texttt{3.16e1}$ for $\texttt{mushrooms}$; $\beta = \texttt{3.73e-1}$ for $\texttt{covtype}$.

For CVA we sweep $p \in [\texttt{1e-3}, \texttt{1e6}]$ and set $\tau = \frac{1}{\| A \|/p + L}$ and $\sigma = \frac{1}{p \| A \|}$ — by construction, these stepsizes satisfy the validity condition and are as large as possible since the condition is satisfied with equality. We do an additional tuning procedure where we choose constants $\tau \in [\texttt{1e-10}, \texttt{1e2}]$ and $\xi \in [\texttt{1e-5}, \texttt{1e2}]$ and set $\sigma = \tau \xi$, which are subject to verifying the stepsize validity condition of CVA. Finally we select the best stepsizes across the two tuning phases to be (truncated to 3 decimals): $\tau = \texttt{9.869e-4}, \sigma = \texttt{1.125e1}$ for $\texttt{ijcnn}$; $\tau = \texttt{2.655e-4}, \sigma = \texttt{7.896e1}$ for $\texttt{a9a}$; $\tau = \texttt{9.936e-4}, \sigma = \texttt{5.878e0}$ for $\texttt{mushrooms}$; $\tau = \texttt{7.728e-06}, \sigma = \texttt{1e-06}$ for $\texttt{covtype}$.

Note that the Hessian of $f$ is given by $\nabla^2 f(x) = Q^T D(x) Q$, where $D(x)$ is a diagonal matrix such that $D_{i,i}(x) = \sigma_i(x)(1 - \sigma_i(x))$, where $\sigma_i(x) = \frac{1}{1 + \exp(-b_i \langle q_i, x \rangle)} \in (0, 1)$. Clearly, over any compact set in $\mathcal{C} \subset \mathcal{X}$ there exist $D_{\min} := \min_{i, x \in \mathcal{C}} D_{i,i}(x) \in (0, 1)$ such that $D_{\min} Q^T Q \preceq Q^T D(x) Q$. As a result, a sufficient condition for local strong convexity is that the minimum eigenvalue of $Q^T Q$ be greater than 0.

The convergence results along with stepsize comparison plots are presented in Figure 1. For dataset $\texttt{ijcnn}$ we run APDA with the modified $\tau_k$ used in Theorem 4.2, since $\lambda_{\min}(Q^T Q) = 75.13$ and $A$ has full rank. In the latter case, the legend identifier is $\texttt{APDA-strcnv}$. For the remaining datasets we use only the basic setting for $\tau_k$, as $\lambda_{\min}(Q^T Q) \leq \texttt{1e-13}$.

While APDA outperforms FISTA and CVA on $\texttt{ijcnn}$, $\texttt{a9a}$ and $\texttt{mushrooms}$, it shows a relatively poor performance on $\texttt{covtype}$. We hypothesize that this is related to the condition number of $Q^T Q$, which is almost three orders of magnitude larger in the latter case: $\texttt{9.2e22}$ versus $\texttt{5.3e1}$, $\texttt{2e20}$ and $\texttt{2e17}$ for $\texttt{ijcnn}$, $\texttt{mushrooms}$ and $\texttt{a9a}$, respectively. A similar behavior is seen in Figure 1.(c) of [Malitsky and Mishchenko, 2020].

Finally, the adaptive property of APDA's stepsizes is visible in the stepsize comparison plots where they are shown to oscillate within at least one order of magnitude throughout the optimization process.

## 6 Limitations of APDA

The experiments presented in this paper (Section 5 and Appendix A) have the common trait of not imposing hard constraints on the primal variables. As a consequence, we are able to take plain gradient steps in the primal domain. However, for instances such as Poisson linear inverse problems [Bertero et al., 2009], the iterates $x_k$ need to reside in $\mathbb{R}^n_+$ because the primal objective contains $\log$ functions. APDA cannot handle such cases, as any constraints imposed on the primal variables will only be satisfied asymptotically. We consider such scenarios as a future research direction.

## Acknowledgements

The first author is grateful to Ya-Ping Hsieh for his feedback on the manuscript and for helpful research discussions throughout the development of this work. The authors also sincerely thank the anonymous reviewers for their time and their thoughtful, constructive feedback which helped improve and clarify this manuscript.

This project has received funding from the European Research Council (ERC) under the European Union's Horizon 2020 research and innovation programme (grant agreement no. 725594 - time-data), the Wallenberg AI, Autonomous Systems and Software Program (WASP) funded by the Knut and Alice Wallenberg Foundation, with the project number 305286. The work was also sponsored by the Department of the Navy, Office of Naval Research (ONR) under a grant number N62909-17-1-2111; by the Army Research Office and was accomplished under Grant Number W911NF-19-1-0404; by the Hasler Foundation Program: Cyber Human Systems (project number 16066). This work was also supported by the Swiss National Science Foundation (SNSF) under grant number 200021_178865/1.

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

# Appendix

## A  Additional experiments

### A.1  Nonconvex phase retrieval

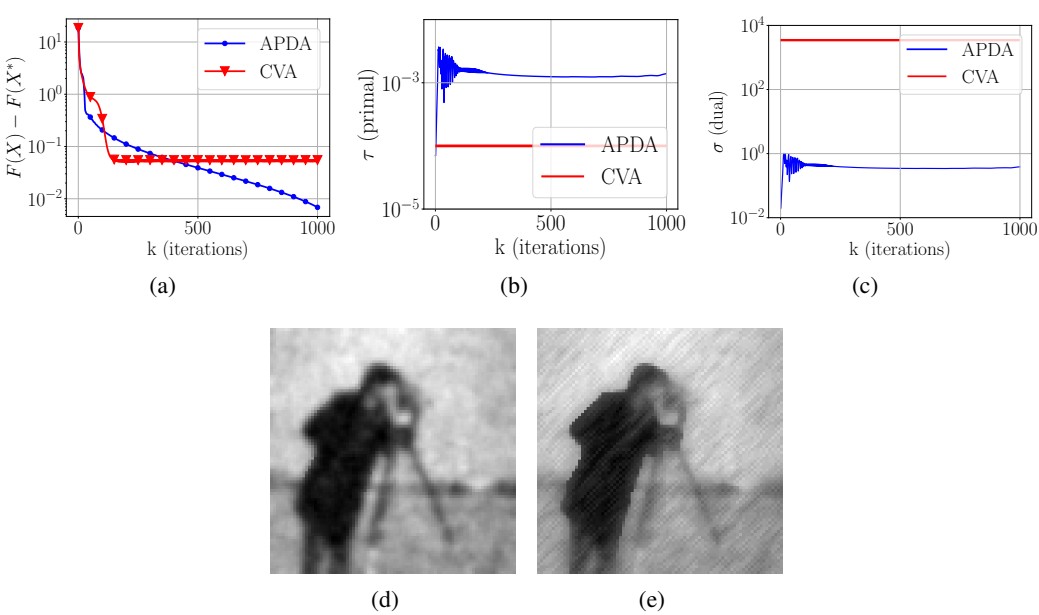

(a)

(b)

(c)

(d)

(e)

Figure 2: (a) Convergence rate. (b) Primal stepsize comparison. (c) Primal stepsize comparison. (d) APDA reconstruction, PSNR = 21.34, SSIM = 0.76. (e) CVA reconstruction, PSNR = 20.56, SSIM = 0.70.

In this section we provide the results for applying our algorithm, heuristically, on the nonconvex least squares formulation of the phase retrieval (PR) problem. The phase-retrieval problem has attracted intense interest recently, due to its application is domains such as optical imaging [Walther, 1963], astronomy [Fienup and Dainty, 1987] and many others. Here, we consider the real counterpart of the original complex PR formulation for square images, where given $\{(A_i, b_i) \in \mathbb{R}^{n \times n} \times \mathbb{R}\}$ we want to recover $X \in \mathbb{R}^{n \times n}$ up to its sign, such that $b_i = \mathrm{Tr}(A_i^T x)^2$. To this end, we consider the following optimization objective:

$$\min_{X \in \mathbb{R}^{n \times n}} F(X) := \underbrace{\frac{1}{4m} \sum_{i=1}^{m} \left(b_i - \mathrm{Tr}(A_i^T X)^2\right)^2}_{f(X)} + \underbrace{\lambda \|DX\|_{2,1}}_{g(X) \equiv \|\cdot\|_{\mathrm{TV}}}. \tag{10}$$

We note a few things: first, objective (10) is nonconvex with $f$ being only locally smooth. Secondly, Sun et al. [2018] have recently shown that given $m$ i.i.d Gaussian measurements, the global geometry of $F(X)$ is 'benign' for $m > Cd \log(d)^3$, where $d$ is the problem dimension. By benign, the authors specifically mean '(1) there are no spurious local minimizers, and all global minimizers are equal to the target signal $x$ up to a global phase; and (2) the objective function has a negative directional curvature around each saddle point'. It is posed that in such cases iterative algorithms should, with high probability, find the minimizer without requiring special initialization as is needed for current state of the art solvers.

For our experiments we use $84 \times 84$-sized images and choose a smaller number of measurements than suggested above: $m = d \log(d) \approx 27,155$. We generate $m$ sparse matrices $A_i \in R^{n \times n}$ with $30\%$ non-zero entries sampled i.i.d from the standard normal distribution, and corrupt a random subset containing $10\%$ of elements in $b_i$ by setting them to 0. We perform parameter sweep for $\lambda \in [\texttt{1e-4}, \texttt{1e4}], \beta \in [\texttt{1e-3}, \texttt{1e4}]$ and settle for $\lambda = \texttt{1e2}$ and $\beta = \texttt{2.78e2}$. Without guidelines for setting $\tau, \sigma$ for CVA since $f$ is not $L$-smooth, we search for the best $\tau \in [\texttt{1e-4}, \texttt{1e4}]$ and

$p \in [\text{1e-2}, \text{1e2}]$ such that $\sigma = \frac{1}{p\tau\|A\|}$ and settle for $\tau = \text{1e-4}$, $p = \text{1.02}$. We note that CVA diverged for 32/40 grid points, whereas our method converged for all instances. Finally, the initial points $x_0$ and $y_0$ are sampled from the standard normal distribution.

The results are depicted in Figure 2, which contains the reconstructions and convergence plots. For each reconstruction we report the Peak Signal to Noise Ratio (PSNR) and the Structural Similarity Index Measure (SSIM). We tested several random seeds and obtained similar results. We also tried running CVA with the stepsize values used by APDA in its last iteration (notice how in Figure 2 (d) $\tau_k$ essentially stabilizes in a very narrow band just above 1e-3 after the first 250 iterations) — however, CVA diverged in this setting as well.

## A.2 Image inpainting

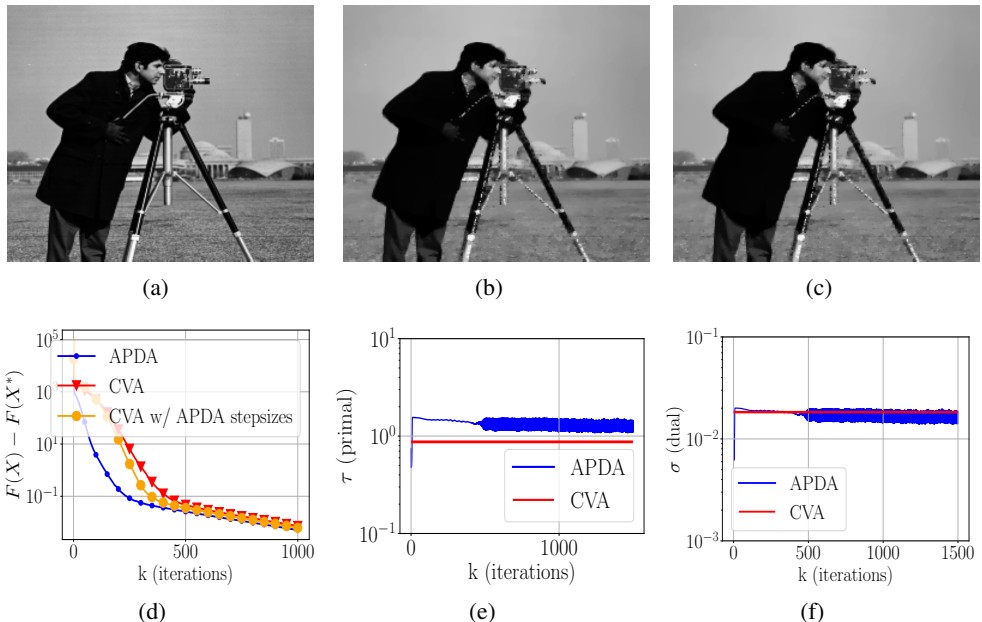

Figure 3: (a) Original image downloaded from http://www.cs.tut.fi/~foi/GCF-BM3D/. (b) APDA reconstruction, PSNR = 25.63, SSIM = 0.91. (c) CVA reconstruction, PSNR = 25.63, SSIM = 0.91. (d) Convergence rate. (e) Primal stepsize comparison. (f) Dual stepsize comparison.

Image inpainting consists in reconstructing the missing parts of a subsampled image $b = P_\Omega(X^\natural)$, where $P_\Omega : \mathbb{R}^{m \times n} \to \mathbb{R}^{m \times n}$ is an operator that selects a subset of $q$ pixels from the original image $X^\natural \in \mathbb{R}^{m \times n}$, where $q \ll mn$. This problem can be formulated as a regularized optimization objective:

$$\min_{X \in \mathbb{R}^{m \times n}} F(X) := \underbrace{\frac{1}{2}\|b - P_\Omega(X)\|_2^2}_{f(X)} + \underbrace{\lambda\|DX\|_{2,1}}_{g(X)\equiv\|\cdot\|_{\text{TV}}}, \tag{11}$$

where $D : \mathbb{R}^{m \times n} \to \mathbb{R}^{m \times n \times 2}$ represents the discrete gradient operator, and $\|DX\|_{2,1} = \sum_{i,j=1}^{m,n} \sqrt{(DX)_{i,j,1}^2 + (DX)_{i,j,2}^2}$. The regularization term represents the isotropic TV norm, which is known to help in recovering sharp images by preserving discontinuities and reducing noise [Chambolle et al., 2010, Condat, 2017].

For our experiments, we vectorize the images of size $256 \times 256$ and transform $D$ accordingly. We represent $P_\Omega$ as a matrix built by removing rows uniformly at random from $I$ and which removes 60% of pixels from the original image (sampling ratio 0.4). We perform parameter sweep for $\lambda \in [\text{1e-4}, \text{1e}]$, and settle for $\lambda = \text{1e-2}$. We also sweep $\beta \in [\text{1e-5}, \text{1e}]$ and settle for $\beta =$

`1.291e-2`. Finally, we perform a similar two-phase tuning for CVA as that described in Section 5 with $p \in [\texttt{1e-5}, \texttt{1e3}]$ for the first phase and $\tau \in [\texttt{1e-5}, \texttt{1e2}]$, $\xi \in [\texttt{1e-5}, \texttt{1e1}]$ for the second phase. We settle for stepsizes $\tau = \texttt{8.722e-1}$ and $\sigma = \texttt{1.831e-2}$.

Experiment results are presented in Figure 3, where we show the reconstructions, alongside the convergence plot and a comparison of the fixed stepsizes of CVA with those of APDA. The two algorithms are comparable both in terms of reconstruction quality and convergence speed, with APDA being marginally better for the latter criterion. The convergence plot also shows an instance of CVA whose stepsizes were set to the values of those used by APDA in the final iteration of these experiments. Finally, subfigures (e) and (f) show APDA's stepsizes oscillating within close range of CVA's.

# B  Missing proofs

## B.1  Proof of Lemma 4.1

**Lemma 4.1.** *Consider APDA along with Assumptions 3.1 and 3.2 and $(x, y) \in \mathcal{X} \times \mathcal{Y}$. Then, for all $k$ and $\eta_k \in \left( \frac{\beta \tau_k \| A \|}{1 - c}, \frac{1 - 2\tau_k L_k}{2\tau_k \| A \|} \right)$,*

$$\| x_{k+1} - x \|^2 + \frac{1}{\beta} \| y_{k+1} - y \|^2 + (1 - \eta_k \tau_k \| A \| - \tau_k L_k) \| x_{k+1} - x_k \|^2$$

$$+ \frac{\eta_k - \tau_k \beta \| A \|}{\beta \eta_k} \| y_{k+1} - y_k \|^2 + 2\tau_k(1 + \theta_k) P_{x,y}(x_k) + 2\tau_k D_{x,y}(y_{k+1})$$

$$\leq \| x_k - x \|^2 + \frac{1}{\beta} \| y_k - y \|^2 + \tau_k L_k \| x_k - x_{k-1} \|^2 + 2\tau_k \theta_k P_{x,y}(x_{k-1}).$$

*Moreover, it holds that:*

*1) $\tau_k L_k < \frac{1}{2} < 1 - \eta_k \tau_k \| A \| - \tau_k L_k$,*

*2) $\frac{1}{\beta} - \frac{\tau_k \| A \|}{\eta_k} > \frac{c}{\beta} > 0$.*

*Proof.* Using the primal update rule, we have

$$\| x_{k+1} - x \|^2 = \| x_k - x \|^2 + \| x_{k+1} - x_k \|^2 - 2\tau_k \langle \nabla f(x_k) + A^T y_{k+1}, x_k - x \rangle. \quad (12)$$

We address each term in the RHS separately. Using the convexity of $f$ we bound the last term of (12):

$$-2\tau_k \langle \nabla f(x_k) + A^T y_{k+1}, x_k - x \rangle \leq 2\tau_k \left( f(x) - f(x_k) \right) + 2\tau_k \langle A(x - x_k), y_{k+1} \rangle. \quad (13)$$

For the second term of (12) we use an expansion similar to the analysis in [Malitsky and Mishchenko, 2020] along with the primal update rule:

$$\| x_{k+1} - x_k \|^2 = 2 \| x_{k+1} - x_k \|^2 - \| x_{k+1} - x_k \|^2$$

$$= 2\tau_k \langle \nabla f(x_k) + A^T y_{k+1}, x_k - x_{k+1} \rangle - \| x_{k+1} - x_k \|^2$$

$$= 2\tau_k \langle \nabla f(x_k) - \nabla f(x_{k-1}), x_k - x_{k+1} \rangle + 2\tau_k \langle A^T y_{k+1} - A^T y_k, x_k - x_{k+1} \rangle$$

$$+ 2\tau_k \langle \nabla f(x_{k-1}) + A^T y_k, x_k - x_{k+1} \rangle - \| x_{k+1} - x_k \|^2. \quad (14)$$

Notice that the first term in (14) gives us the opportunity to insert a dependence on the local Lipschitz constant $L_k$. Using Cauchy-Schwarz, the definition of $L_k$ and Young's inequality, we indeed take this opportunity and get:

$$\langle \nabla f(x_k) - \nabla f(x_{k-1}), x_k - x_{k+1} \rangle \leq L_k \| x_k - x_{k-1} \| \| x_{k+1} - x_k \|$$

$$\leq \frac{L_k}{2} \left( \| x_k - x_{k-1} \|^2 + \| x_{k+1} - x_k \|^2 \right). \quad (15)$$

Similarly, we bound the second term in (14) and obtain:

$$\langle A^T y_{k+1} - A^T y_k, x_k - x_{k+1} \rangle \leq \frac{\|A\| \eta}{2} \|x_{k+1} - x_k\|^2 + \frac{\|A\|}{2\eta} \|y_{k+1} - y_k\|^2, \qquad (16)$$

where $\eta > 0$ is a free parameter coming from Young's inequality.

Finally, for the third term in (14) we use the update rule and the convexity of $f$:

$$\langle \nabla f(x_{k-1}) + A^T y_k, x_k - x_{k+1} \rangle = \langle \frac{1}{\tau_{k-1}}(x_{k-1} - x_k), \tau_k(\nabla f(x_k) + A^T y_{k+1}) \rangle$$
$$\leq \theta_k \left( f(x_{k-1}) - f(x_k) \right) + \theta_k \langle A(x_{k-1} - x_k), y_{k+1} \rangle. \qquad (17)$$

Replacing (15), (16) and (17) into (14), we get

$$\|x_{k+1} - x_k\|^2 \leq \tau_k L_k \|x_k - x_{k-1}\|^2 + (\tau_k \|A\| \eta + \tau_k L_k - 1) \|x_{k+1} - x_k\|^2$$
$$+ \frac{\tau_k \|A\|}{\eta} \|y_{k+1} - y_k\|^2 + 2\tau_k \theta_k \left( f(x_{k-1}) - f(x_k) \right)$$
$$+ 2\tau_k \theta_k \langle A(x_{k-1} - x_k), y_{k+1} \rangle. \qquad (18)$$

Finally, replacing (18) and (13) back into (12) and using the fact that $\theta_k \langle A(x_{k-1} - x_k), y_{k+1} \rangle + \langle A(x - x_k), y_{k+1} \rangle = -\langle A(\tilde{x}_k - x), y_{k+1} \rangle$, we obtain the inequality for the primal iterate sequence:

$$\|x_{k+1} - x\|^2 \leq \|x_k - x\|^2 + \tau_k L_k \|x_k - x_{k-1}\|^2 + (\tau_k \|A\| \eta + \tau_k L_k - 1) \|x_{k+1} - x_k\|^2$$
$$+ \frac{\tau_k \|A\|}{\eta} \|y_{k+1} - y_k\|^2 + 2\tau_k \theta_k \left( f(x_{k-1}) - f(x_k) \right) + 2\tau_k \left( f(x) - f(x_k) \right)$$
$$- 2\tau_k \langle A(\tilde{x}_k - x), y_{k+1} \rangle. \qquad (19)$$

We now seek a similar result for the dual sequence. For this, we use the following characterization of the proximal operator:

$$u = \operatorname{prox}_{g^*}(x) \iff \langle u - x, z - u \rangle \geq g^*(u) - g^*(z) \; \forall z. \qquad (20)$$

Thus, letting $u = y_{k+1}$, $x = y_k$ and $z = y$ in (20), we obtain:

$$g^*(y) \geq g^*(y_{k+1}) + \langle \frac{1}{\sigma_k}(y_k - y_{k+1}), y - y_{k+1} \rangle + \langle A\tilde{x}_k, y - y_{k+1} \rangle.$$

Using the cosine rule for the second term, the fact that $\sigma_k = \beta\tau_k$ and multiplying both sides by $2\tau_k > 0$, we obtain:

$$\frac{1}{\beta} \|y_{k+1} - y\|^2 \leq \frac{1}{\beta} \|y_k - y\|^2 - \frac{1}{\beta} \|y_{k+1} - y_k\|^2 + 2\tau_k(g^*(y) - g^*(y_{k+1}))$$
$$+ 2\tau_k \langle A\tilde{x}_k, y_{k+1} - y \rangle. \qquad (21)$$

Summing (21) with (19) we obtain the following recurrence:

$$\|x_{k+1} - x\|^2 + \frac{1}{\beta} \|y_{k+1} - y\|^2$$
$$\leq \|x_k - x\|^2 + \frac{1}{\beta} \|y_k - y\|^2 + \tau_k L_k \|x_k - x_{k-1}\|^2$$
$$+ (\tau_k \|A\| \eta + \tau_k L_k - 1) \|x_{k+1} - x_k\|^2 + \left( \frac{\tau_k \|A\|}{\eta} - \frac{1}{\beta} \right) \|y_{k+1} - y_k\|^2$$
$$+ 2\tau_k \left( \theta_k \left( f(x_{k-1}) - f(x_k) \right) + f(x) - f(x_k) + g^*(y) - g^*(y_{k+1}) \right.$$
$$+ 2\tau_k \langle Ax, y_{k+1} \rangle - 2\tau_k \langle A\tilde{x}_k, y \rangle. \qquad (22)$$

We further process the terms involving $f$ and $g^*$ on the right-hand side in order to form the $P_{x,y}(\cdot)$ and $D_{x,y}(\cdot)$:

$$f(x) - f(x_k) = -P_{x,y}(x_k) + \langle A(x_k - x), y \rangle,$$

$$\theta_k(f(x_{k-1}) - f(x_k)) = \theta_k P_{x,y}(x_{k-1}) - \theta_k P_{x,y}(x_k) + \langle \theta_k A(x_k - x_{k-1}), y \rangle,$$

$$g^*(y) - g^*(y_{k+1}) = -D_{x,y}(y_{k+1}) - \langle Ax, y_{k+1} - y \rangle.$$

Replacing the above expressions into (22) and noting that $\langle A(\tilde{x}_k - x), y \rangle - \langle Ax, y_{k+1} - y \rangle + \langle Ax, y_{k+1} \rangle - \langle A\tilde{x}_k, y \rangle = 0$, we obtain:

$$\| x_{k+1} - x \|^2 + \frac{1}{\beta} \| y_{k+1} - y \|^2 + (1 - \tau_k \| A \| \eta - \tau_k L_k) \| x_{k+1} - x_k \|^2$$

$$+ \left( \frac{1}{\beta} - \frac{\tau_k \| A \|}{\eta} \right) \| y_{k+1} - y_k \|^2 + 2\tau_k(1 + \theta_k) P_{x,y}(x_k) + 2\tau_k D_{x,y}(y_{k+1})$$

$$\leq \| x_k - x \|^2 + \frac{1}{\beta} \| y_k - y \|^2 + \tau_k L_k \| x_k - x_{k-1} \|^2 + 2\tau_k \theta_k P_{x,y}(x_{k-1}). \quad (23)$$

What is left to do for obtaining the stated result is to choose $\eta$, possibly depending on $k$, such that the corresponding terms are positive. First, note that $\tau_k L_k \| x_k - x_{k-1} \|^2 < \frac{1}{2} \| x_k - x_{k-1} \|^2$ because $z \mapsto \frac{z}{2\sqrt{z^2+a}}$, $a > 0$ is an increasing function whose limit at $\infty$ is $\frac{1}{2}$ and we have:

$$\tau_k L_k \leq \frac{L_k}{2\sqrt{L_k^2 + (\beta/(1-c)) \| A \|^2}} < \frac{1}{2}. \quad (24)$$

Next we need to choose $\eta = \eta_k$ (iteration-dependent) to satisfy:

$$\begin{cases} \frac{1}{\beta} - \frac{\tau_k \| A \|}{\eta_k} > 0, \\ 1 - \tau_k \| A \| \eta_k - \tau_k L_k > \frac{1}{2}. \end{cases}$$

However, for theoretical purposes related to controlling the sequence $\| y_{k+1} - y_k \|^2$, we strengthen the first inequality to $\frac{1}{\beta} - \frac{\tau_k \| A \|}{\eta_k} > \frac{c}{\beta}$, $c \in (0,1)$. In practice, this constant is chosen as small as possible. The new conditions to be satisfied are:

$$\begin{cases} \frac{1}{\beta} - \frac{\tau_k \| A \|}{\eta_k} > \frac{c}{\beta}, \\ 1 - \tau_k \| A \| \eta_k - \tau_k L_k > \frac{1}{2}, \end{cases} \iff \begin{cases} \eta_k > \frac{\beta \tau_k \| A \|}{1-c}, \\ \eta_k < \frac{1 - 2\tau_k L_k}{2\tau_k \| A \|}. \end{cases} \quad (25)$$

The question we need to answer therefore is: given the expression of $\tau_k$, is the interval always valid for choosing $\eta_k \in \left( \frac{\beta \tau_k \| A \|}{1-c}, \frac{1 - 2\tau_k L_k}{2\tau_k \| A \|} \right)$?

To answer, we form the corresponding quadratic inequality in $\tau_k$:

$$\frac{\beta \tau_k \| A \|}{1-c} - \frac{1 - 2\tau_k L_k}{2\tau_k \| A \|} < 0 \iff \frac{2\beta \tau_k^2 \| A \|^2}{1-c} + 2\tau_k L_k - 1 < 0, \quad (26)$$

whose 2 real roots are given by:

$$\begin{cases} \tau_{k,1} = \frac{1}{L_k - \sqrt{L_k^2 + 2(\beta/(1-c)) \| A \|^2}} & < 0, \\ \tau_{k,2} = \frac{1}{L_k + \sqrt{L_k^2 + 2(\beta/(1-c)) \| A \|^2}} & > 0. \end{cases}$$

For inequality (26) to be satisfied, we need:

$$\tau_k \in (0, \tau_{k,2}) = \left( 0, \frac{1}{L_k + \sqrt{L_k^2 + 2(\beta/(1-c)) \| A \|^2}} \right), \forall k. \quad (27)$$

The lower bound for $\tau_k$ trivially holds, and for the upper bound we make the following observation:

$$L_k + \sqrt{L_k^2 + 2(\beta/(1-c))\left\|A\right\|^2} = \frac{2\left[\sqrt{L_k^2} + \sqrt{L_k^2 + 2(\beta/(1-c))\left\|A\right\|^2}\right]}{2}$$

$$\overset{\text{Jensen}}{<} 2\sqrt{\frac{2L_k^2 + 2(\beta/(1-c))\left\|A\right\|^2}{2}}$$

$$= 2\sqrt{L_k^2 + (\beta/(1-c))\left\|A\right\|^2}.$$

Here Jensen's inequality holds strictly because function $\sqrt{\cdot}$ is strictly concave and $L_k^2 \neq L_k^2 + 2\left\|A\right\|^2 \beta$. Thus, we obtain:

$$0 < \tau_k \leq \frac{1}{2\sqrt{L_k^2 + \left\|A\right\|^2 \beta}} < \frac{1}{L_k + \sqrt{L_k^2 + 2\beta\left\|A\right\|^2}} = \tau_{k,2} \quad \forall k.$$

It follows that we can find an $\eta_k \in \left(\dfrac{\beta\tau_k\left\|A\right\|}{1-c}, \dfrac{1-2\tau_k L_k}{2\tau_k\left\|A\right\|}\right)$, $\forall k$, which implies that conditions (25) can always be satisfied. This concludes the proof. $\qquad\square$

## B.2 Proof of Theorem 4.1

**Theorem 4.1.** *Consider APDA along with Assumptions 3.1 and 3.2, and let $(x^*, y^*) \in \mathcal{X} \times \mathcal{Y}$ be a saddle point of problem (2). Then, for all $k$*

1) ***Boundedness.*** *The sequence $\{(x_k, y_k)\}$ is bounded. Specifically, for all $k$,*

$$\left\|x_k - x^*\right\|^2 + \left\|y_k - y^*\right\|^2 \leq M,$$

*where $M := \left\|x_1 - x^*\right\|^2 + \frac{1}{\beta}\left\|y_1 - y^*\right\|^2 + \frac{1}{2}\left\|x_1 - x_0\right\|^2 < \infty$.*

2) ***Convergence to a saddle point.*** *The sequence $\{(x_k, y_k)\}$ converges to a saddle point of (2).*

3) ***Ergodic convergence.*** *Let $S_k := \sum_{i=1}^{k}\tau_i$, $X_k := \dfrac{1}{S_k}\left(\tau_k(1 + \theta_k)x_k + \sum_{i=1}^{k-1}\left(\tau_i(1 + \theta_i) - \tau_{i+1}\theta_{i+1}\right)x_i\right)$ and $Y_k := \dfrac{1}{S_k}\sum_{i=1}^{k}\tau_i y_{i+1}$. Then, for any bounded $B_1 \times B_2 \in \mathcal{X} \times \mathcal{Y}$ and for all $k$,*

$$\mathcal{G}_{B_1 \times B_2}(X_k, Y_k) \leq \frac{M(B_1, B_2)\sqrt{L^2 + (\beta/(1-c))\left\|A\right\|^2}}{k},$$

*where $L$ is the Lipschitz constant of $\nabla f$ over the compact set $\overline{\text{Conv}}(\{x^*, x_0, x_1, \ldots\})$ and $M(B_1, B_2) = \sup_{(x,y) \in B_1 \times B_2}\left\|x_1 - x\right\|^2 + \frac{1}{\beta}\left\|y_1 - x\right\|^2 + \frac{1}{2}\left\|x_1 - x_0\right\|^2$.*

*Proof.* **1) Sequence boundedness.** Using the inequality of Lemma (4.1) with $(x, y) = (x^*, y^*)$ and the fact that $\tau_k L_k < \frac{1}{2}$, $\forall k$, unrolling it over the iterations and rearranging the terms we obtain:

$$\left\|x_{k+1} - x^*\right\|^2 + \frac{1}{\beta}\left\|y_{k+1} - y^*\right\|^2 + (1 - \eta_k\tau_k\left\|A\right\| - \tau_k L_k)\left\|x_{k+1} - x_k\right\|^2$$

$$+ \sum_{i=1}^{k-1}\left(\frac{1}{2} - \eta_i\tau_i\left\|A\right\| - \tau_i L_i\right)\left\|x_{i+1} - x_i\right\|^2 + \frac{c}{\beta}\sum_{i=1}^{k}\left\|y_{i+1} - y_i\right\|^2 + 2\tau_k(1 + \theta_k)P_{x^*,y^*}(x_k)$$

$$+ 2\sum_{i=2}^{k-1}\left(\tau_i(1 + \theta_i) - \tau_{i+1}\theta_{i+1}\right)P_{x^*,y^*}(x_i) + 2\sum_{i=1}^{k}\tau_i D_{x^*,y^*}(y_{i+1})$$

$$\leq \| x_1 - x^* \|^2 + \frac{1}{\beta} \| y_1 - y^* \|^2 + \frac{1}{2} \| x_1 - x_0 \|^2 + 2\tau_1\theta_1 P_{x^*,y^*}(x_0). \qquad (28)$$

All the terms on the left hand-side of (28) are non-negative:

$$\begin{cases} \dfrac{1}{\beta} - \dfrac{\tau_k \| A \|}{\eta_i} > \dfrac{c}{\beta} > 0, \ \forall i, & \text{(by Lemma 4.1)} \\[2mm] \dfrac{1}{2} - \eta_i\tau_i \| A \| - \tau_i L_i > 0, \ \forall i, & \text{(by Lemma 4.1)} \\[2mm] \tau_{i+1}\theta_{i+1} \leq \tau_i\sqrt{1+\theta_i}\ \theta_{i+1} \leq \tau_i(1+\theta_i), & \text{(by stepsize update rule)} \\[2mm] P_{x^*,y^*}(x) \geq 0, \ D_{x^*,y^*}(y) \geq 0 \ \forall x, y. & \text{(by the saddle point property)} \end{cases}$$

Also, by our parameter setup we have that $\theta_1 = 0$. Consequently, it holds that:

$$\| x_{k+1} - x^* \|^2 + \frac{1}{\beta} \| y_{k+1} - y^* \|^2 \leq M < \infty \ \forall k,$$

where $M := \| x_1 - x^* \|^2 + \frac{1}{\beta} \| y_1 - y^* \|^2 + \frac{1}{2} \| x_1 - x_0 \|^2$, which implies that the sequence is bounded.

We make the following remarks which will be useful for the remainder of the theorem's proof:

- Boundedness of $\{x_k\}$ together with the local Lipschitz continuity of $\nabla f$ from Assumption 3.1 implies that there exists $L > 0$ such that $f$ is $L$-smooth over $\overline{\mathrm{Conv}}(\{x^*, x_0, x_1, \ldots\})$. Furthermore, $L \geq L_k \ \forall k$.

- A consequence of the prior point is that $\tau_k$ has a uniform and positive lower-bound. By the definition of APDA it holds that:

$$\tau_1 = \frac{1}{2\sqrt{L_1^2 + (\beta/(1-c)) \| A \|^2}} \geq \frac{1}{2\sqrt{L^2 + (\beta/(1-c)) \| A \|^2}}$$

and, from the definition of $\tau_k$ it is straightforward to see that at every iteration we either explicitly increase $\tau_k$ relative to $\tau_{k-1}$ or otherwise set it to an expression dictated by the local smoothness constant $L_k$. Thus it holds that:

$$\tau_k \geq \frac{1}{2\sqrt{L^2 + (\beta/(1-c)) \| A \|^2}}, \ \forall k. \qquad (29)$$

- Furthermore, the existence of $L$ guarantees that $\tau_k L_k$ can have a tighter upper bound than the $1/2$ shown before, as follows:

$$\tau_k L_k \leq \frac{L_k}{2\sqrt{L_k^2 + (\beta/(1-c)) \| A \|^2}}$$

$$\leq \frac{L}{2\sqrt{L^2 + (\beta/(1-c)) \| A \|^2}}, . \qquad (30)$$

where we used the fact that $z \mapsto \frac{z}{2\sqrt{z^2+a}}$, $a > 0$ is an increasing function.

- Finally, due to the point above, we can uniformly lower bound the coefficients of terms $\| x_{k+1} - x_k \|^2$ on the LHS of (28), and thus obtain:

$$\frac{1}{2}\left( 1 - \frac{L}{\sqrt{L^2 + (\beta/(1-c)) \| A \|^2}} \right) \sum_{i=1}^{k-1} \| x_i - x_{i+1} \|^2 + \frac{c}{\beta} \sum_{i=1}^{k} \| y_{i+1} - y_i \|^2 \leq M,$$

which conveniently ensures that:

$$\begin{cases} \lim\limits_{k\to\infty} \| x_k - x_{k-1} \|^2 = 0, \\[2mm] \lim\limits_{k\to\infty} \| y_k - y_{k-1} \|^2 = 0. \end{cases} \qquad (31)$$

**2) Convergence to a saddle point.** Let $(\hat{x}, \hat{y})$ be an arbitrary cluster point of the sequence $\{(x_k, y_k)\}$. Since we have shown that the sequence is bounded, then there must exist a subsequence $\{(x_{k_i}, y_{k_i})\}$, such that $\lim_{i \to \infty}(x_{k_i}, y_{k_i}) = (\hat{x}, \hat{y})$. We wish to prove that $(\hat{x}, \hat{y})$ is a saddle point of (2).

More precisely, we wish to prove that $P_{\hat{x}, \hat{y}}(x) \geq 0$ and $D_{\hat{x}, \hat{y}}(y) \geq 0$ for $\forall x, y$, respectively. For convenience, we remind the reader the definitions of these two quantities:

$$P_{\hat{x}, \hat{y}}(x) = f(x) - f(\hat{x}) + \langle A(x - \hat{x}), \hat{y} \rangle,$$
$$D_{\hat{x}, \hat{y}}(y) = g^*(y) - g^*(\hat{y}) - \langle A\hat{x}, y - \hat{y} \rangle.$$

We start with $P_{\hat{x}, \hat{y}}(x)$:

$$
\begin{aligned}
P_{\hat{x}, \hat{y}}(x) &= f(x) - f(\hat{x}) + \langle A(x - \hat{x}), \hat{y} \rangle \\
&= \lim_{i \to \infty} f(x) - f(x_{k_i}) + \langle A(x - x_{k_i}), y_{k_i} \rangle && \text{(Continuity of } f) \\
&\geq \lim_{i \to \infty} \langle \nabla f(x_{k_i}) + A^* y_{k_i+1}, x - x_{k_i} \rangle + \langle A^*(y_{k_i} - y_{k_i+1}), x - x_{k_i} \rangle && \text{(Convexity of } f) \\
&= \lim_{i \to \infty} \langle \frac{x_{k_i+1} - x_{k_i}}{\tau_{k_i}}, x - x_{k_i} \rangle + \langle A^*(y_{k_i} - y_{k_i+1}), x - x_{k_i} \rangle && \text{(Primal update rule)} \\
&= 0. && \text{(By (29), (31))}
\end{aligned}
$$

Showing the analogous result for $D_{\hat{x}, \hat{y}}(y)$ relies on similar arguments, with the additional requirement that $\theta_k$ is uniformly upper bounded. From the update rule of $\tau_k$ we have:

$$\theta_k = \frac{\tau_k}{\tau_{k-1}} \leq \sqrt{1 + \theta_{k-1}} \implies \theta_k \leq \sqrt{1 + \ldots + \sqrt{1 + \theta_2}} \leq \underbrace{\sqrt{1 + \ldots + \sqrt{1 + 1}}}_{k-2 \text{ times}} \leq 2, \quad (32)$$

where the second to last inequality comes from the way APDA's first two iterations are set up.

Therefore, we have that $\forall y \in \mathcal{Y}$:

$$
\begin{aligned}
D_{\hat{x}, \hat{y}}(y) &= g^*(y) - g^*(\hat{y}) - \langle A\hat{x}, y - \hat{y} \rangle \\
&\geq g^*(y) - \liminf_{i \to \infty} g^*(y_{k_i}) - \langle A \liminf_{i \to \infty} x_{k_i}, y - \liminf_{i \to \infty} y_{k_i} \rangle && \text{(l.s.c. of } g^*) \\
&= \limsup_{i \to \infty} g^*(y) - g^*(y_{k_i}) - \langle A x_{k_i}, y - y_{k_i} \rangle \\
&\geq \limsup_{i \to \infty} \langle \frac{y_{k_i-1} - y_{k_i}}{\sigma_{k_i-1}}, y - y_{k_i} \rangle + \langle A(\tilde{x}_{k_i-1} - x_{k_i}), y - y_{k_i} \rangle && \text{(Poperty (20))} \\
&= \limsup_{i \to \infty} \langle \frac{y_{k_i-1} - y_{k_i}}{\beta \tau_{k_i-1}}, y - y_{k_i} \rangle + \langle A\left[x_{k_i-1} - x_{k_i} + \theta_{k_i-1}(x_{k_i-1} - x_{k_i-2})\right], y - y_{k_i} \rangle \\
&= 0. && \text{(By (29), (31), (32))}
\end{aligned}
$$

**3) Gap rate.** Unrolling the inequality of Lemma 4.1 for some $(x, y) \in B_1 \times B_2$, we obtain:

$$
\| x_{k+1} - x \|^2 + \frac{1}{\beta} \| y_{k+1} - x \|^2 + (1 - \eta_k \tau_k \| A \| - \tau_k L_k) \| x_{k+1} - x_k \|^2
$$

$$
+ \sum_{i=1}^{k-1} \left( \frac{1}{2} - \eta_i \tau_i \| A \| - \tau_i L_i \right) \| x_{i+1} - x_i \|^2 + \frac{c}{\beta} \sum_{i=1}^{k} \| y_{i+1} - y_i \|^2 + 2\tau_k (1 + \theta_k) P_{x,y}(x_k)
$$

$$
+ 2 \sum_{i=2}^{k-1} \left( \tau_i (1 + \theta_i) - \tau_{i+1} \theta_{i+1} \right) P_{x,y}(x_i) + 2 \sum_{i=1}^{k} \tau_i D_{x,y}(y_{i+1})
$$

$$
\leq \| x_1 - x \|^2 + \frac{1}{\beta} \| y_1 - y \|^2 + \frac{1}{2} \| x_1 - x_0 \|^2. \quad (33)
$$

First, note that due to $\theta_1 = 0$, the following holds:

$$\tau_k (1 + \theta_k) + \sum_{i=1}^{k-1} \left( \tau_i (1 + \theta_i) - \tau_{i+1} \theta_{i+1} \right) = \sum_{i=1}^{k} \tau_i =: S_k.$$

Second, since all the terms on the LHS of (33) except those involving $P_{x,y}(\cdot)$ and $D_{x,y}(\cdot)$ are non-negative and, for fixed $(x, y) \in \mathcal{X} \times \mathcal{Y}$ the functions $P_{x,y}(\cdot)$ and $D_{x,y}(\cdot)$ are convex, we have:

$$2S_k\left(P_{x,y}(X_k) + D_{x,y}(Y_k)\right)$$
$$\leq 2\tau_k(1+\theta_k)P_{x,y}(x_k) + 2\sum_{i=2}^{k-1}\left(\tau_i(1+\theta_i) - \tau_{i+1}\theta_{i+1}\right)P_{x,y}(x_i) + 2\sum_{i=1}^{k}\tau_i D_{x,y}(y_{i+1})$$
$$\leq \|\,x_1 - x\,\|^2 + \frac{1}{\beta}\|\,y_1 - y\,\|^2 + \frac{1}{2}\|\,x_1 - x_0\,\|^2. \tag{34}$$

Lastly, since $\tau_k \geq \frac{1}{2\sqrt{L^2 + (\beta/(1-c))\|\,A\,\|^2}}$, $\forall k$, we have that $S_k \geq \frac{k}{2\sqrt{L^2 + (\beta/(1-c))\|\,A\,\|^2}}$ and the rate for the restricted gap is:

$$\mathcal{G}_{B_1 \times B_2}(X_k, Y_k)$$
$$= \sup_{(x,y)\in B_1 \times B_2} P_{x,y}(X_k) + D_{x,y}(Y_k)$$
$$\leq \sup_{(x,y)\in B_1 \times B_2} \frac{\left(\|\,x_1 - x\,\|^2 + \frac{1}{\beta}\|\,y_1 - y\,\|^2 + \frac{1}{2}\|\,x_1 - x_0\,\|^2\right)\sqrt{L^2 + (\beta/(1-c))\|\,A\,\|^2}}{k}$$
$$= \frac{M(B_1, B_2)\sqrt{L^2 + (\beta/(1-c))\|\,A\,\|^2}}{k},$$

which concludes the proof of the theorem. $\qquad\square$

### B.3 Proof of Theorem 4.2

Before proving the result of Theorem 4.2, a few remarks are in order. First, the boundedness result of Theorem 4.1 point 1) also holds for constant $c = 0$, since this constant was required only for proving convergence to a saddle point in point 2) of the theorem. Second, taking a stepsize smaller than the originally considered $\tau_k$ will not change the validity of Lemma 4.1 or the boundedness result of Theorem 4.1, as it remains within the interval given in (27).

Consequently, for studying APDA under the additional Assumption 3.3 we can simplify the stepsize expression by taking $c = 0$, since now we will prove convergence of the iterates directly by using the strong convexity and full row-rank assumptions. Specifically, we consider $\tau_k$ as defined in (8), which is smaller than the one originally considered and, due to the above remarks it ensures that APDA produces a bounded sequence. It follows that, under the local smoothness and local strong convexity assumptions, there exist constant $L$ and $\mu$ such that $f$ is $L$-smooth and $\mu$-strongly convex over $\overline{\text{Conv}}(\{x^*, x_0, x_1, \ldots\})$. This observation suffices to show linear convergence in Theorem 4.2.

**Theorem 4.2.** *Consider APDA along with Assumptions 3.1, 3.2 and 3.3. Let $(x^*, y^*) \in \mathcal{X} \times \mathcal{Y}$ be a saddle point of problem (2). Furthermore, let $\tau_k$ be defined by (8) and let $s := \sqrt{4L^2 + \beta\|\,A\,\|^2}$ and $t := \sqrt{4\mu^2 + \beta\|\,A\,\|^2}$, where $\mu$, $L$ are the strong convexity and smoothness constants of $f$ over the compact set $\overline{\text{Conv}}(\{x^*, x_0, x_1, \ldots\})$.*

*Then, for all $k$:*
$$\|\,x_k - x^*\,\|^2 + \frac{1}{\beta}\|\,y_k - y^*\,\|^2 \leq (1 - \min\{p, q, r\})^k M,$$

*where the rate constants are given by:*
$$p = \frac{1}{2}, \quad q = \frac{\mu}{4s}, \quad r = \frac{\beta\sigma_{\min}^2(A)\mu}{\beta\sigma_{\min}^2(A)\mu + 8s^2t + 4L^2s},$$

*and* $M = \|x_2 - x^*\|^2 + \left(\frac{1}{\beta} + T\right)\|y_2 - y^*\|^2 + \frac{1}{2}\|x_2 - x_1\|^2 + 2\tau_1 P_{x^*,y^*}(x_1)$, $T = \frac{\sigma_{\min}^2(A)\mu}{8s^2t + 4L^2s}$, *with* $\sigma_{\min}(A)$ *representing the smallest singular value of A.*

*Proof.* The outline of the proof is first arriving at a strengthened version of the inequality in Lemma 4.1, and then showing that the inequality expresses a contraction.

Since this new stepsize still ensures the boundedness result of Theorem 4.1, there exist $\mu$ and $L$ such that $f$ is $\mu$-strongly convex and $L$-Lipschitz smooth over the compact set $\overline{\text{Conv}}(\{x^*, x_0, x_1, \ldots\})$. From these properties it follows that, for all $k$:

$$2\tau_k\langle\nabla f(x_k), x^* - x_k\rangle \leq 2\tau_k\left(f(x^*) - f(x_k)\right) - \mu\tau_k\|x_k - x^*\|^2,$$
$$2\tau_k\langle\nabla f(x_k), x^* - x_k\rangle \leq 2\tau_k\left(f(x^*) - f(x_k)\right) - \frac{\tau_k}{L}\|\nabla f(x_k) - \nabla f(x^*)\|^2.$$

Summing these two inequalities and dividing by 2, we obtain a stronger version of equation (13):

$$-2\tau_k\langle\nabla f(x_k) + A^T y_{k+1}, x_k - x^*\rangle \leq 2\tau_k\left(f(x^*) - f(x_k)\right) - \frac{\tau_k\mu}{2}\|x_k - x^*\|^2$$
$$- \frac{\tau_k}{2L}\|\nabla f(x_k) - \nabla f(x^*)\|^2 + 2\tau_k\langle A(x^* - x_k), y_{k+1}\rangle. \quad (35)$$

We further bound the term $\|\nabla f(x_k) - \nabla f(x^*)\|^2$ in (40):

$$\|\nabla f(x^*) - \nabla f(x_k)\|^2 = \left\|A^T(y_{k+1} - y^*) - \frac{x_k - x_{k+1}}{\tau_k}\right\|^2 \quad (36)$$
$$\geq \|A^T(y_{k+1} - y^*)\|^2 + \frac{1}{\tau_k^2}\|x_{k+1} - x_k\|^2$$
$$- \frac{2}{\tau_k}\|A^T(y_{k+1} - y^*)\|\|x_{k+1} - x_k\| \quad (37)$$
$$\geq \|A^T(y_{k+1} - y^*)\|^2 + \frac{1}{\tau_k^2}\|x_{k+1} - x_k\|^2$$
$$- \left(\frac{1}{\xi+1}\|A^T(y_{k+1} - y^*)\|^2 + \frac{\xi+1}{\tau_k^2}\|x_{k+1} - x_k\|^2\right) \quad (38)$$
$$\geq \frac{\xi\sigma_{\min}^2(A)}{\xi+1}\|y_{k+1} - y^*\|^2 - \frac{\xi}{\tau_k^2}\|x_{k+1} - x_k\|^2, \quad (39)$$

where line (36) comes from the primal iterate update rule and the optimality condition (4); line (37) comes from developing the square and applying Cauchy-Schwarz; line (38) comes from applying Young's inequality with constant $1 + \xi$, where $\xi > 0$; line (39) comes from the assumption of $A$ having full-row rank, which implies that $\|A^T(y_{k+1} - y^*)\|^2 \geq \sigma_{\min}^2(A)\|y_{k+1} - y^*\|^2$.

Finally, setting $\xi = 2\tau_k^2 L_k L$ we obtain that:

$$-2\tau_k\langle\nabla f(x_k) + A^T y_{k+1}, x_k - x^*\rangle \leq 2\tau_k\left(f(x^*) - f(x_k)\right) - \frac{\tau_k\mu}{2}\|x_k - x^*\|^2$$
$$- \frac{\tau_k^3 L_k \sigma_{\min}^2(A)}{1 + 2\tau_k^2 L_k L}\|y_{k+1} - y^*\|^2 + \tau_k L_k\|x_{k+1} - x_k\|^2$$
$$+ 2\tau_k\langle A(x^* - x_k), y_{k+1}\rangle. \quad (40)$$

Replacing inequality (13) with inequality (40) in the proof of Lemma 4.1 and keeping everything else identical, we obtain the a strengthened version of Lemma's 4.1 result:

$$\|x_{k+1} - x^*\|^2 + \left(\frac{1}{\beta} + \frac{\tau_k^3 L_k \sigma_{\min}^2(A)}{1 + 2\tau_k^2 L_k L}\right)\|y_{k+1} - y^*\|^2 + \left(1 - \eta_k\tau_k\|A\| - 2\tau_k L_k\right)$$

$$\|x_{k+1} - x_k\|^2 + \frac{\eta_k - \tau_k\beta\|A\|}{\beta\eta_k}\|y_{k+1} - y_k\|^2 + 2\tau_k(1 + \theta_k)P_{x^*,y^*}(x_k) + 2\tau_k D_{x^*,y^*}(y_{k+1})$$

$$\leq \left(1 - \frac{\mu \tau_k}{2}\right) \| x_k - x^* \|^2 + \frac{1}{\beta} \| y_k - y^* \|^2 + \tau_k L_k \| x_k - x_{k-1} \|^2 + 2\tau_k \theta_k P_{x^*, y^*}(x_{k-1}). \quad (41)$$

In order to show that this is in fact a contraction, we note a few properties of the terms in (41):

a) It holds that $1 - \eta_k \tau_k \| A \| - 2\tau_k L_k > 1/2$ and $\frac{\eta_k - \tau_k \beta \| A \|}{\beta \eta_k} > 0$ since:

$$\begin{cases} 1 - \eta_k \tau_k \| A \| - 2\tau_k L_k > 1/2 \\ \frac{\eta_k - \tau_k \beta \| A \|}{\beta \eta_k} > 0 \end{cases} \iff \begin{cases} \eta_k < \frac{1}{2\tau_k \| A \|} - \frac{2L_k}{\| A \|} \\ \eta_k > \tau_k \beta \| A \| \end{cases}$$

$$\iff 2\beta \| A \|^2 \tau_k^2 + 4L_k \tau_k - 1 < 0,$$

which holds for any $\tau_k \in \left(0, \dfrac{1}{2L_k + \sqrt{4L_k^2 + 2\beta \| A \|^2}}\right)$. Our choice of $\tau_k$ belongs to this interval, and therefore ensures the stated properties;

b) It holds that $\tau_k L_k < 1/4$, by the same observation as that in (24) but with a different limit constant given by the new stepsize;

c) It holds that $\tau_k \theta_k \leq \tau_{k-1}\sqrt{1 + \theta_{k-1}/2}\,\theta_k \leq \tau_{k-1}(1 + \theta_{k-1}/2)$, by the definitions of $\tau_k$ and $\theta_k$;

d) It holds that:

$$\frac{1}{2\sqrt{4L^2 + \beta \| A \|^2}} \leq \tau_k \leq \frac{1}{2\sqrt{4\mu^2 + \beta \| A \|^2}}, \quad (42)$$

by the existence of $\mu$ and $L$ over $\overline{\mathrm{Conv}}(\{x^*, x_0, x_1, \dots\})$ and a similar argument to that in (29), plus the fact that under strong convexity $\| \nabla f(x) - \nabla f(y) \| \geq \mu \| x - y \|$;

e) It holds that:

$$\frac{\mu}{2\sqrt{4\mu^2 + \beta \| A \|^2}} \leq \tau_k L_k \leq \frac{L}{2\sqrt{4L^2 + \beta \| A \|^2}}, \quad (43)$$

by a similar argument to that in (30).

Using properties a), b), c) in the list above and ignoring the positive terms on the LHS that do not have a correspondent on the RHS of (41), the main inequality becomes:

$$\| x_{k+1} - x^* \|^2 + \left(\frac{1}{\beta} + T\right) \| y_{k+1} - y^* \|^2 + \frac{1}{2} \| x_{k+1} - x_k \|^2 + 2\tau_k(1 + \theta_k) P_{x^*, y^*}(x_k)$$

$$\leq \left(1 - \frac{\mu \tau_k}{2}\right) \| x_k - x^* \|^2 + \frac{1}{\beta} \| y_k - y^* \|^2 + \frac{1}{2}\left(1 - \frac{1}{2}\right) \| x_k - x_{k-1} \|^2$$

$$+ 2\tau_{k-1}(1 + \theta_{k-1}/2) P_{x^*, y^*}(x_{k-1}), \quad (44)$$

where $T$ is given by:

$$T := \frac{\sigma_{\min}^2(A)\mu}{8s^2 t + 4L^2 s}$$

$$= \frac{\sigma_{\min}^2(A)\mu}{8(4L^2 + \beta \| A \|^2)\sqrt{4\mu^2 + \beta \| A \|^2} + 4L^2\sqrt{4L^2 + \beta \| A \|^2}}$$

$$\leq \frac{\tau_k^3 L_k \sigma_{\min}^2(A)}{1 + 2\tau_k^2 L_k L}, \quad \text{(by d) and e) above)}$$

where we used the definitions of $s = \sqrt{4L^2 + \beta \| A \|^2}$ and $t = \sqrt{4\mu^2 + \beta \| A \|^2}$ to simplify notations.

We thus have the following contractions in (44):

- For $\frac{1}{2}\left\Vert x_{k+1}-x_k\right\Vert^2$ it is: $1-\underbrace{\dfrac{1}{2}}_{=:p}$ ;

- For $\left(\frac{1}{\beta}+T\right)\left\Vert y_{k+1}-y^*\right\Vert^2$ it is:

$$
\begin{aligned}
1+\frac{1}{1+T\beta} &= 1-\frac{T\beta}{1+T\beta} \\
&= 1-\underbrace{\frac{\beta\sigma_{\min}^2(A)\mu}{\sigma_{\min}^2(A)\mu+8s^2t+4L^2s}}_{=:r}
\end{aligned}
$$

- For $\left\Vert x_{k+1}-x^*\right\Vert^2$ it is:

$$
1-\frac{\mu\tau_k}{2}\leq 1-\underbrace{\frac{\mu}{4s}}_{=:q}
$$

- For $2\tau_k(1+\theta_k)P_{x^*,y^*}(x_k)$ it is:

$$
\begin{aligned}
\frac{1+\theta_{k-1}/2}{1+\theta_{k-1}} &= 1-\frac{\theta_{k-1}}{2(1+\theta_{k-1})} \\
&\leq 1-\frac{t}{s} \qquad\qquad \text{(By def. of } \theta_{k-1} \text{ and property 5.)}
\end{aligned}
$$

Note that for the latter two contractions above, it always holds that $\mu/(4s)<t/s$ so in the final bound we can ignore the latter. Finally, denoting the LHS of inequality (44) as $E_{k+1}$, we have that:

$$
E_{k+1}\leq\left(1-\min\{p,q,r\}\right)^{k+1}M.
$$

where $M=E_2$ and we used the fact that $\theta_1=0$. $\qquad\square$