# OpenReview forum: "A first-order primal-dual method with adaptivity to local smoothness"
_NeurIPS.cc/2021/Conference — NeurIPS 2021 Poster_

### Official Review · Reviewer_RYzj · 2021-07-09

**Rating:** 6
**Confidence:** 4

**Summary:**

Interesting contributions in the field of optimization but not well enough presented and poor match for Neurips.

**Limitations And Societal Impact:**

Yes, this is a theoretical work

**Main Review:**

The paper presents interesting contributions in convex optimization. But since this is an optimization paper and the experiments are in image processing, without any relationship to machine learning, I think the paper would be more suitable in an optimization journal.

Independently, I think that the state of the art should be presented in a better way: to minimize f(x) + g(Ax) using the gradient of f and the prox. of g, there are 2 main algorithms: 1) the Condat-Vu algorithm (CVA) and 2) an algorithm proposed independently by several authors and called the PAPC, PDFP2O or Loris-Verhoeven algorithm (LVA), see e.g. in the paper
Condat et al., "Proximal splitting algorithms: A tour of recent advances, with new twists".
The CVA should not be called PDHG = the Chambolle Pock algorithm, since this is confusing: this algorithm for this problem would use the prox. of f. The abstract, in particular, is misleading.
The LVA has been proved to converge linearly when f is strongly convex and A is full rank; I don't know if this is the case for the CVA.
So, I understand that your algorithm is an adaptive extension of the CVA. It should be presented as such and the properties discussed: in the globally Lipschitz case, do you recover what is known about the CVA?

In Figure 1, it seems that tau and sigma converge almost immediately to some values, around 0.2 and 0.02. I guess that if you use these values in the CVA, you get the same convergence profile. So, CVA is slower only because you chose the wrong stepsize values. So, you method is a way to select automatically the stepsize values in the CVA, which is nice and useful in practice. This is what should be advertised. The "local adaptivity" property is not illustrated; in particular, as mentioned at the end of the paper, the method is not suitable for barrier function which are not Lipschitz continuous.

I don't think the experiment in Section 5.2 is relevant, since your theory is in the convex case, so application to a particular nonconvex example does not show anything.

=== update after author response period
The responses to the different concerns raised by the reviewers are convincing. My initial evaluation was perhaps too severe: it is true that automatic selection of the stepsizes in optimization algorithms is a difficult but very important topic, so every progress in this area is welcome.
And contrary to other reviewers, I don't think that the work is a marginal step after the work of Malitsky and Mishchenko, since primal-dual methods need particular care to be tamed.
So, assuming that the authors will implement the changes they promise to do, I am increasing my score from 4 to 6.


**Time Spent Reviewing:**

1

---

> ### Author Response · Authors · 2021-08-10
> **Thank you for the feedback and suggestions**
>
> We thank the reviewer for their time, their thoughtful comments and the helpful suggestions for improving our paper. We address the major concerns point by point below.
>
> * **Regarding our work not being relevant to machine learning:** We agree that the main-text applications could have been chosen to better illustrate the relevance of our paper to the ML community. To this end, we propose to swap the phase-retrieval experiments of Section 5.2 with the sparse logistic regression ones in Appendix A.2, since the latter is inarguably of interest to the ML community.
>
>     In addition, we note that for $A = I$ our template also covers problem formulations of the kind $\min_x f(x) + g(x)$, which are usually solved by proximal  gradient schemes. Thus the "for-free" adaptivity and relaxed smoothness conditions can also be extended to these methods, which cover many problem formulations in ML (see e.g. [2]).
>
>     Finally, since our algorithm can be seen as an enhancement of CVA which plays an important role in modern optimization,  we argue that our work fits into the NeurIPS call for papers: “we invite submissions [...] on topics including but not limited to the following: [...] Optimization (e.g., convex and non-convex optimization)”.
>
> * **Regarding the naming PDHG/CVA:** We understand that our naming was not appropriate and misrepresents the Condat-Vu algorithm. We apologise and will rectify this in the revision.
>
> * **Regarding the literature review \& recovering what is known for CVA:**  Firstly, we thank the reviewer for pointing out the gaps in our presentation of the literature, which we will address accordingly in the revision.
>
>     Secondly, regarding recovering what is known for CVA for the global smoothness case, the theoretical rate for solving our template with $L$-smooth $f$ in [3] is given by:
> $$\mathcal{G}_{x, y}(x_N, y_N) \leq \frac{1}{N}\left(\frac{1}{2\tau} \lVert x - x_0\rVert^2 + \frac{1}{2\sigma} \lVert y - y_0\rVert^2 + 2 \langle A(x -x_0) , (y - y_0) \rangle\right).$$
>
>   In order to make it (as) comparable (as possible) to our rate, we upper bound the inner product term using Cauchy-Schwarz and Young's inequality for $\alpha > 0$, to get:
> $$\mathcal{G}_{x, y}(x_N, y_N) \leq \frac{1}{N}\left(\left(\frac{1}{2\tau} + \lVert A\rVert\alpha\right)\lVert x - x_0\rVert^2 + \left(\frac{1}{2\sigma} + \frac{\lVert A\rVert}{\alpha}\right)\lVert y - y_0\rVert^2 \right).$$
>
>    Considering  that $\tau,\sigma$ need to satisfy the stability  condition in line (145)  of our paper and  are usually set as $\tau = \frac{1}{\lVert A  \rVert /p + L_f}$ $\sigma = \frac{1}{p\lVert A \rVert}$ for some appropriate $p >0$ (line 146 of paper), we have  that
> $$\mathcal{G}_{x, y}(x_N, y_N) \leq \frac{1}{N}\left(\left( 2L_f + 2\left(\frac{1}{p} + \alpha \right)\lVert A \rVert \right)\lVert x - x_0 \rVert^2 + 2\left(p+\frac{1}{\alpha}\right)\lVert A\rVert \lVert y - y_0 \rVert^2 \right).$$
>
>    For APDA, noting that $\theta_1 = 0$ (coming from setting  $\tau_0 = \infty$) and  upper bounding the square root term using $\sqrt{a_1} + \sqrt{a_2} \geq \sqrt{a_1 + a_2}$, the rate of Thm 4.1 can be written as:
>
>   $$ \mathcal{G}_{x, y}(x_N, y_N) \leq \frac{\left(L_\mathrm{conv} + \sqrt{\beta/(1-c)} \lVert A \rVert \right)}{N}  \left( \lVert x - x_0 \rVert^2 + \frac{1}{\beta}\lVert y - y_0\rVert^2 + \lVert x_1 - x_0 \rVert^2\right)$$
>
>   The results written in this format make it easier to see that we achieve the same rate. The constants are not directly comparable, but the quantities they involve bear the same meaning (smoothness constants with $L_{\mathrm{conv}} \leq L_f$, $\lVert A \rVert$ and initial distance to optimum). While our worst-case bounds may be slightly worse, APDA can take consistently larger stepsizes than the baseline ( Fig. 1, 2 ) and can converge faster in practice.
>
>   Finally, indeed, linear convergence results for CVA under assumptions identical to ours don't seem to be published (linear rates are usually shown for all 3 operators in CVA being non-zero, without assumptions on A -- see e.g. [3]). However, in the case of $L$-smooth and $\mu$-strongly-convex $f$ and full row-rank $A$ (the same setup as ours except the former two conditions hold locally for APDA), [4] shows the linear convergence of PDFP$^2$O for solving our problem formulation with rate
> $$
> \lVert x_k - x^*\rVert^2  \leq \left(\lVert x_1 - x_0 \rVert^2 + \frac{1}{\sigma_{\mathrm{max}}(A)} \lVert y_1 - y_0\rVert^2\right) \left( 1 - \min \left( \frac{\sigma_{\mathrm{min}}(A)}{\sigma_{\mathrm{max}}(A)}, \frac{\mu}{L}\right)\right)^{k-1}.
> $$
>    In this case, a direct comparison of the constants is even less achievable than the previous case, given that the expression of our bound is much more involved (which is also why we do not restate it here). It is safe to assume the constants in our bound are worse, which is inevitable given the kind of analysis required under our weaker regularity assumptions and for achieving this style of adaptivity.
>
> * **Regarding using our method for estimating CVA stepsizes:**
> Unfortunately, our method cannot provide good estimates for CVA for 2 reasons: (1) While the stepsizes indeed seem to stabilize towards the end of the optimization process, there is no theoretical guarantee that this should happen; as such, it is not clear how these values could be chosen in a principled way. (2) APDA’s $\tau_k$ and $\sigma_k$ are not guaranteed to stabilize to valid values of CVA stepsizes. Specifically, CVA requires that $\tau$ and $\sigma$ verify a global stability condition (line 145 of the paper), in order for its theoretical guarantees to hold. Using stepsizes outside of this condition would turn CVA into a heuristic. We illustrate this with the specific problem of Section 5.1: the stepsizes used by APDA in its final iteration (part of the ‘stable’ region) are $\tau \simeq 1.697$ and $\sigma \simeq 0.016$, while the other problem constants are $L_f  = 1$ and $\lVert A \rVert \simeq 2.828$. These values unfortunately do not satisfy the stepsize requirements of CVA. We also emphasize that the CVA stepsizes in the experiments were tuned for best convergence in accordance to the recommended procedure mentioned in line (146) of our paper (also see lines 267-270 for details of the tuning, as well as the code provided in the supplementary submission).
>
>   We hope that these clarifications shed more light on the benefits of APDA: it abolishes the need to verify a global stepsize condition, and can choose consistently larger stepsizes compared to CVA by adapting to the local geometry “on the fly”.
>
> * **Regarding the local adaptivity property not being illustrated:** The local stepsize adaptivity property is illustrated by the oscillatory behaviour of the blue curve in the stepsize plots of Fig. 1 and 2. In particular, the blue curve visibly oscillates for almost the entire optimization process in Fig.1 as a result of the local geometry-based rule used for determining the stepsize.
>
> * **Regarding the experiment in section 5.2 not being relevant:** Indeed, the application of APDA to the phase retrieval problem is a heuristic one. However, we argue that our choice is not baseless, as it relies on the interesting results of [1] who show that for an appropriately large number of random measurements, the function $f$ satisfies: "(1) there are no spurious local minimizers, and all global minimizers are equal to the target signal $x$, up to a global phase; and (2) the objective function has a negative directional
> curvature around each saddle point". Moreover, $f$ is only locally smooth, thus fitting our assumption. The authors of [1] perform an experiment on such a problem instance and show that GD with random initialization consistently returns a global minimizer. Our experiment reproduces the observation in [1] and shows that APDA can work even outside of its theoretical guarantees -- an observation which may be the basis for a future research direction.
>
>   Nevertheless, we would be happy to swap the experiments in section 5.2 with those in Appendix A.2 addressing sparse logistic regression.
>
>
>
> [1] Ju Sun, Qing Qu, and John Wright. A geometric analysis of phase retrieval.
>
> [2]Polson, N. G., Scott, J. G., Willard, B. T., and others. “Proximal algorithms in statistics and machine learning.”
>
> [3] Antonin Chambolle and Thomas Pock. On the ergodic convergence rates of a first-order primal–dual algorithm.
>
> [4] Chen, P., Huang, J., Zhang, X.: A primal–dual fixed point algorithm for convex separable minimization with applications to image restoration.

---

> > ### Author Response · Authors · 2021-08-20
> > **Additional experimental data**
> >
> > We follow  up with some additional information about the experiments and provide plots which are accessible through the following link:
> >
> > https://imgur.com/a/RrU8PNa
> >
> >
> > Figure 1. in the link illustrates an experiment in which we run CVA with the stepsizes in the stabilized region of APDA for the problem of Section 5.1. We note that "CVA-orig" denotes the same CVA instance as that used in the paper. The CVA instance using the APDA stepsizes can be seen to diverge -- as mentioned in our previous reply these stepsizes are not compliant with CVA requirements.
> >
> > However, we further tested the idea of running CVA with APDA stepsizes in a different scenario where the reviewer's suspicion is indeed confirmed. Setting $\beta = \texttt{7.74263683e-02}$ in APDA yields stepsizes that stabilize to a value that is correct for CVA. We ran CVA with these compliant stepsizes and observed better convergence. In addition, we re-tuned CVA with more parameters over a 2D grid, instead of relying on the possibly restrictive scheme described in lines 146-147 of the paper (we "free-tuned"). We used 70 points for this grid-search and thus provide a stronger baseline than before. The results are provided in Figure 2. from the link.
> >
> > The observations we can thus make are:
> >   * Running CVA with the stepsizes of APDA can  make it converge comparably fast, should the  stepsizes be compliant with the stability condition;
> >   * CVA potentially needs a large number of points to be tuned for good performance.
> >
> > We'd be happy to include these observations in the revision and thank the reviewer for pointing us in this direction.

---

### Official Review · Reviewer_nTWw · 2021-07-14

**Rating:** 7
**Confidence:** 5

**Summary:**

This paper studies a saddle-point problem with bilinear coupling and no constraints on primal as well as dual variables. Due to the unboundedness of the underlying problem, the paper assumes local Lipschitz smoothness for any compact sets. It proposes an adaptive version of the primal-dual method that estimates the local Lipschitz smoothness constant of the primal function. The authors show that the iterates remain in a bounded set around the saddle-point. Consequently, they can show asymptotic convergence of the limit point. They also show a convergence rate for a weaker notion of the primal-dual gap.

**Limitations And Societal Impact:**

While I agree with most of the analysis, there is one thing I have to mention about the convergence criterion primal-dual gap: P(X) + D(Y).
As defined in this paper, if the problem is \min_x \max_y F(x,y) then primal-dual gap P(X) +D(Y) = F(X, y^*) - F(x^*, Y). This is a weaker criterion than \sup_{x \in \XX,  y \in \YY} F(X, y) - F(x, Y) for some compact set \XX, \YY containing saddle-point (x^*, y^*).
To see this, consider a simple case of F(x, y) = y^TAx. Clearly, the unconstrained saddle-point problem has solution (x^*, y^*) = (0,0).
In that case, P(X) + D(Y) = 0 trivially for all X, Y. However, the supremum-based criterion above is 0 iff (X, Y) is a saddle-point. It might be worth checking whether the gap convergence can be proved for the supremum-based criterion.

An update after the response:
I am satisfied with the responses by the authors and keep my score.

**Main Review:**

The novel part of this paper has some novelty in the analysis showing the boundedness of primal and dual variables. It improves upon the setting of Malitsky et al. [2018]. Compared to Malitsky et al. [2018], they assume that the primal function does not have an easy proximal evaluation oracle. To estimate the local Lipschitz constant of the primal function, they use the recently proposed approximation technique by Malitsky and Mishchenko [2020].

The paper provides a clear exposition of the ideas and gives thorough proof of all claims. The content is easy to comprehend and an experienced reader can grasp the details quickly.

An issue in the proof of Theorem 4.1 part (3):
I cannot see the relation \tau_k(1+\theta_k) + \sum_{i = 1}^{k-1} (\tau_i(1+\theta_i) - \tau_{i+1}\theta_{i+1}) = \sum_{i = 1}^k \tau_i. I am getting an additional \tau_1\theta_1. Please double-check on whether I am missing anything. If not then please fix this issue.

**Time Spent Reviewing:**

7 hours

---

> ### Author Response · Authors · 2021-08-10
> **Thank you for the feedback and suggestions**
>
> We thank the reviewer for their time, their insightful comments and the helpful suggestions for improving our paper. We address the comments point by point below.
>
> * **Issue in the proof of Thm. 4.1:** We thank  the reviewer for pointing this out -- indeed, we did not explain this statement sufficiently. The reason why the term $\tau_1\theta_1$ does not appear is because $\theta_1= 0$. This comes from the initial value of $\tau_0 = \infty$ and the fact that $\theta_1 = \tau_1/\tau_0$, with $\tau_1 < \infty$. We will clarify this aspect.
>
>
> * **Proving convergence for the supremum-based criterion:**  We are very grateful to the reviewer for this insightful observation. It seems to us that we indeed could show this, as the result of Lemma 4.1 can hold for arbitrary $x \in \mathcal{X}$ and $y \in \mathcal{Y}$ (not just $x^*, y^*$). The unrolled version of the result over the  iterations (corresponding to eq. (36) in Appendix) will have as the right-hand side $\lVert x_1 - x \rVert^2 + \frac{1}{\beta}\lVert y_1 - y\rVert^2 + \tau_1L_1\lVert x_1 - x_0 \rVert^2$ (using again the fact that $\theta_1 = 0$). Consequently, we could lower-bound the LHS of eq. (36), where all the non-gap terms are non-negative, by using Jensen and forming $\mathcal{G}_{x, y}(X_k, Y_k)$ to arrive at the result. We will amend the revision accordingly and we thank the reviewer again for pointing this out.

---

> > ### Comment · Reviewer_nTWw · 2021-08-31
> > **Response**
> >
> > Authors have responded positively to my comments as well as sufficiently addressed concerns of other reviewers. They have engaged in fruitful discussion. I am satisfied with overall paper provided that they make the suggested changes in the final version. I keep my score.

---

### Official Review · Reviewer_prmj · 2021-07-16

**Rating:** 6
**Confidence:** 3

**Summary:**

The authors propose an adaptive stepsize updating rule for the primal-dual Chambolle-Pock method that solves min f(x)+g(Ax) for locally-smooth f and cheap-prox-step g. The updating rule for adaptive stepsizes is simple in the sense that it only uses local gradients and ||A||, without needing any further subroutines like line search. They provide theoretical convergence rate for their proposed method under the local-smooth assumption and also with further local-strongly-convex assumption. They also corroborated their findings with experiment results that shows favorable  behavior of the new step-sizes.

**Limitations And Societal Impact:**

Yes, the authors included the limitation of their work in Sec 5.3.

**Main Review:**

The paper consists some interesting ideas towards improving the practicality of primal-dual methods. The method it proposes is new to primal-dual methods for minimax problem and leads to good empirical performance. My main concerns on the paper is:

1) the novelty of the method: the method is essentially a generalization of the adaptive stepsize updating rule proposed for SGD in Malitsky and Mishchenko, 2019. Most of the ideas and techniques seem similar to that paper and it will be good if the authors can make their own design and contribution more explicit in the paper.

2) the theoretical performance: If I understand correctly, the theoretical rate that the authors show didn't fully match the state of the art of standard non-accelerated primal-dual methods, even when local smoothness degrades to global smoothness. For instance, in Theorem 4.2, there seems to be at least a (L/mu)^{1+alpha} and kappa(A)^{1+alpha} dependence, for alpha>1. (also is q correct in unit? why not optimizing q for stating the final complexity?)

Given these factors, I think it is a paper with interesting and meaningful result but am slightly worried about if the results included are significant enough to be published in the venue. If the authors have convincing arguments to relieve my worry or can improve their results further, I am happy to change my mind.

* minor comments:
- The paper will read better if it gives detailed definition of the notations it used, i.e.  dom(g), sigma(A), ... etc.
- The proof sketch of Theorem 1 could be organized in a better way; especially the second and third terms that authors refer to in line 193 looks unclear.
- It would be good to add remarks after Theorem 4.2 to explain more clearly in each regime of L/mu, ||A||, which term in the complexity will dominate and what rate one will get from your method.
- The formatting on line 230 could be improved a bit to avoid confusion in minus sign and hyphen.
- line 307: satisfies -> satisfied

**Time Spent Reviewing:**

4

---

> ### Author Response · Authors · 2021-08-10
> **Thank you for the feedback and suggestions**
>
> We thank the reviewer for their time, their thoughtful comments and the helpful suggestions for improving our paper. We address the major concerns point by point below.
>
> * **Regarding our contribution:** We understand the reviewer’s concern and hope to address it in the following.
>
>     Our contribution is to derive the adaptive stepsize schedule for the considered primal-dual problem formulation, while using only the information provided by $\frac{\lVert \nabla f(x_k) - \nabla f (x_{k-1}) \rVert}{ \lVert x_k - x_{k-1} \rVert}$ and $\lVert A \rVert$.
>
>     We would like to emphasize that, although the technical idea indeed originates from [1], the primal-dual structure of our template poses challenges that are beyond the scope of [1]. The crucial difference is, whereas [1] only deals with a single (primal) stepsize $\tau$, our algorithm additionally involves the dual stepsize $\sigma$ and a relaxation parameter $\theta$. Identifying the correct relation between these parameters requires defining and solving a system of nontrivial inequalities (e.g. appendix, from equation (29) onwards), which leads us to a new style of stepsize for primal-dual methods.
>
>
> * **Regarding the significance/impact of the results:**
>     - To our knowledge, APDA is the first primal dual method solving this problem formulation that achieves adaptivity "for-free" (i.e. without the need to perform linesearch or incur any additional significant computational expense).
>
>     - Our template extends the reach of these methods to locally smooth functions $f$ (resp. locally strongly convex), requiring thus a weaker assumption than previous works (global regularity conditions + knowledge of the constants).
>
>     - Importantly, for $A = I$ our template recovers $\min_x f(x) + g(x)$ as the primal problem, a formulation usually addressed via proximal gradient methods and with many applications in ML ([2]). Thus, we extend the benefits of relaxed smoothness assumptions and computationally-cheap (no linesearch) adaptivity also to this class of important problems. We illustrate this with the sparse logistic regression experiments in Appendix A.2. We also note that it is not clear how, or whether the technique of [1] can be extended to the analysis of proximal gradient schemes solving $\min_x f(x) + g(x)$, and hence our method offers a practical alternative.
>
>   Under these considerations, it is our opinion that APDA can bring value for practitioners.
>
>
> * **Regarding the theoretical performance of APDA:** The theoretical rate for solving our template with $L$-smooth $f$ in [3] (which represents our baseline) is given by:
> $$\mathcal{G}_{x, y}(x_N, y_N) \leq \frac{1}{N}\left(\frac{1}{2\tau} \lVert x - x_0\rVert^2 + \frac{1}{2\sigma} \lVert y - y_0\rVert^2 + 2 \langle A(x -x_0) , (y - y_0) \rangle\right).$$
>
>   In order to make it (as) comparable (as possible) to our rate, we upper bound the inner product term using Cauchy-Schwarz and Young's inequality for $\alpha > 0$, to get:
> $$\mathcal{G}_{x, y}(x_N, y_N) \leq \frac{1}{N}\left(\left(\frac{1}{2\tau} + \lVert A\rVert\alpha\right)\lVert x - x_0\rVert^2 + \left(\frac{1}{2\sigma} + \frac{\lVert A\rVert}{\alpha}\right)\lVert y - y_0\rVert^2 \right).$$
>
>    Finally, considering  that $\tau,\sigma$ need to satisfy the stability  condition in line (145)  of our paper and  are usually set as $\tau = \frac{1}{\lVert A  \rVert /p + L_f}$ $\sigma = \frac{1}{p\lVert A \rVert}$ for some appropriate $p >0$ (line 146 of paper), we have  that
> $$\mathcal{G}_{x, y}(x_N, y_N) \leq \frac{1}{N}\left(\left( 2L_f + 2\left(\frac{1}{p} + \alpha \right)\lVert A \rVert \right)\lVert x - x_0 \rVert^2 + 2\left(p+\frac{1}{\alpha}\right)\lVert A\rVert \lVert y - y_0 \rVert^2 \right).$$
>
>    For APDA, noting that $\theta_1 = 0$ (coming from setting  $\tau_0 = \infty$) and  upper bounding the square root term using $\sqrt{a_1} + \sqrt{a_2} \geq \sqrt{a_1 + a_2}$, the rate of Thm 4.1 can be written as:
>
>   $$ \mathcal{G}_{x, y}(x_N, y_N) \leq \frac{\left(L_\mathrm{conv} + \sqrt{\beta/(1-c)} \lVert A \rVert \right)}{N}  \left( \lVert x - x_0 \rVert^2 + \frac{1}{\beta}\lVert y - y_0\rVert^2 + \lVert x_1 - x_0 \rVert^2\right)$$
>
>   The results written in this format make it easier to see that we achieve the same rate. The constants are not directly comparable, but the quantities they involve bear the same meaning (smoothness constants with $L_{\mathrm{conv}} \leq L_f$, $\lVert A \rVert$ and initial distance to optimum). While our worst-case bounds may be slightly worse, APDA can take consistently larger stepsizes than the baseline ( Fig. 1, 2 ) and can converge faster in practice.
>
>    For the case of $L$-smooth and $\mu$-strongly-convex $f$ and full row-rank $A$ (the same setup  as ours except  the former two conditions hold locally for APDA), [4] shows the linear convergence of PDFP$^2$O with rate
> $$
> \lVert x_k - x^*\rVert^2  \leq \left(\lVert x_1 - x_0 \rVert^2 + \frac{1}{\sigma_{\mathrm{max}}(A)} \lVert y_1 - y_0\rVert^2\right) \left( 1 - \min \left( \frac{\sigma_{\mathrm{min}}(A)}{\sigma_{\mathrm{max}}(A)}, \frac{\mu}{L}\right)\right)^{k-1}
> $$
>    In this case, a direct comparison of the constants is even less achievable than the previous case, given that the expression of our bound is much more involved (which is also why we do not restate it here). It is safe to assume the constants in our bound are worse, which is inevitable given the kind of analysis required under our regularity assumptions and for achieving this style of adaptivity.
>
>    Regarding the question about the unit of $q$ in Thm 4.2,  we believe it is correct since $q < 1$. Given that the constants in this bound are directly determined by stepsize-specific conditions such as that of line (563) in the Appendix, optimizing them is not possible without changing the algorithm/analysis. Also, choosing $\beta$ in the bound such that the contraction is optimal is not possible without knowledge of $L$ and $\mu$ (which is not available, since $L$ and $\mu$ are determined over $\overline{\mathrm{conv}}(x^*, x_0, \ldots)$). We hope to have understood correctly the reviewer's questions, and if that's not the case we'd be happy to provide further (or different) clarifications.
>
>
>
> [1] Yura Malitsky and Konstantin Mishchenko. Adaptive gradient descent without descent.
>
> [2] Polson, N. G., Scott, J. G., Willard, B. T., and others. Proximal algorithms in statistics and machine learning.
>
> [3] Antonin Chambolle and Thomas Pock. On the ergodic convergence rates of a first-order primal–dual algorithm.
>
> [4] Chen, P., Huang, J., Zhang, X.: A primal–dual fixed point algorithm for convex separable minimization with applications to image restoration.

---

> > ### Comment · Reviewer_prmj · 2021-08-15
> > **Some follow-up questions:**
> >
> > The contribution you've mentioned about achieving adaptivity for primal-dual setups sounds good to me.
> > My only concern hasn't been addressed is regarding the theoretical performance of APDA:
> > - I didn't doubt for the non-strongly-convex case they match, my only question is - is it true that for the case of $L$-smooth and $\mu$-strongly-convex  and full row-rank $A$, the convergence rate doesn't have match in terms of the order on sigma_max(A)/sigma_min(A), and L/mu? It appears to me that the rate you obtain *in Thm 4.2* can be as large as L^2/sigma_min^2(A), coming from when q  = min(p,q,r), thus it doesn't recover. Am I making a mistake here? What do you mean by inevitable (as for the primal only case: theorem 2 of [1] does recover the standard rate)?
> > - Another way of talking about the unit confusion I am having (maybe I am missing something simple here) is: it seems the definition of p,q,r in Theorem 4.2 will change if  we consider f' = C*f for some constant C, and thus affecting the rate you are claiming for iteration complexity. Is this not true?

---

> > > ### Author Response · Authors · 2021-08-19
> > > **Thank you for the clarifications**
> > >
> > > We thank the reviewer for the additional clarifications regarding constants -- it is now clear to us what was meant. Indeed, we did not optimize the parameters for attaining linear convergence and rather merely focused on showing that it is attained. Also, as the reviewer pointed out, our constants did not scale with the min-max objective that we study. After re-analyzing our proof, we found  that  we can improve it and we thank the reviewer for raising this issue. We now give details about the modifications and improved rates that we can  attain.
> > >
> > >
> > > Our suboptimal constants stemmed from the combined choice of constant $\delta = 2\tau_k^2$ in line 552 of the appendix and
> > > \begin{equation}
> > > \hspace{30mm}\tau_k = \min \Bigg\\{\frac{1}{2 \sqrt{(L_k + 1/L_k)^2 + (\beta/(1-c))\lVert A \rVert^2}}, \tau_{k-1}\sqrt{1+\theta_{k-1}}  \Bigg\\}.
> > > \end{equation}
> > >
> > >
> > > We propose the following changes: Since the only requirement of $\delta$ is that it's positive (line 550), we can change it to  $\delta = 2\tau_k L_k L$. For $\tau_k$ we modify it as follows:
> > > \begin{equation}
> > > \hspace{30mm}\tau_k = \min \Bigg\\{\frac{1}{2\sqrt{4L_k^2 + \beta\lVert A\rVert^2}}, \tau_{k-1}\sqrt{1+\theta_{k-1}/2}\Bigg\\}.
> > > \end{equation}
> > >
> > > The full list of properties associated to the new $\tau_k$ and $\delta$ is:
> > >
> > >   a) $1 -\epsilon_k\tau_k\lVert A \rVert -2\tau_k L_k > 1/2$;
> > >
> > >   b) $\frac{1}{\beta} - \frac{\tau_k\lVert A \rVert}{\epsilon_k} > 0$;
> > >
> > >   c) $\tau_k L_k < 1/4$ (same argument as line 497 with different limit constant);
> > >
> > >   d) $\tau_k\theta_k \leq \tau_{k-1}\sqrt{1 + \theta_{k-1}/2} \\; \theta_k \leq \tau_{k-1}(1 + \theta_{k-1}/2)$ (by def. of $\tau_k$,$\theta_k$);
> > >
> > >   e) $\displaystyle \frac{1}{2\sqrt{4L^2 + \beta\lVert A \rVert^2}} \leq \tau_k \leq \frac{1}{2\sqrt{4\mu^2 + \beta\lVert A \rVert^2}}$ (same argument as lines 562-563);
> > >
> > >    f) $\displaystyle \frac{\mu}{2\sqrt{4\mu^2 + \beta\lVert A \rVert^2}} \leq \tau_kL_k \leq \frac{L}{2\sqrt{4L^2 + \beta\lVert A \rVert^2}}$ (same argument as line 496).
> > >
> > >
> > >
> > > Note that we also removed $(1-c)$ from $\tau_k$, since now we can prove convergence of iterates directly using strong convexity/full rank assumptions. This choice of $\tau_k$ and $\delta$ ensure that we can always choose an $\epsilon_k > 0$ such that points a) and b) in  the list above hold, which together with  c) imply the iterate boundedness result (Thm  4.1, 1)). As a consequence, we then  have the existence of $\mu, L > 0$ such that $f$ is $L$-smooth and $\mu$-strongly convex over $\overline{\mathrm{conv}}( \\{x^*,  x_1, \ldots\\})$ from which points e)  and f) follow.
> > >
> > >
> > >
> > > Using the new $\delta$, eq. (49) in the appendix becomes:
> > >
> > > \begin{align}
> > > \lVert x_{k+1} - x^\*\rVert^2 + \left(\frac{1}{\beta} + \frac{\sigma_{\mathrm{min}}^2(A)\tau_k^3 L_k}{1+2\tau_k^2 L_k L} \right) \lVert y_{k+1} - y^\* \rVert + \Bigg(1 -\epsilon_k\tau_k\lVert A \rVert -2\tau_k L_k\Bigg)
> > > \lVert x_{k+1} - x_k \rVert^2& \\\\
> > > &\hspace{-130mm}+ \left(\frac{1}{\beta}  - \frac{\tau_k\lVert A \rVert }{\epsilon_k} \right)\lVert y_{k+1} - y_k\rVert^2 + 2\tau_k(1+\theta_k)  P_{x^\*, y^\*}(x_{k}) + 2\tau_k D_{x^\*, y^\*}(y_{k+1}) \\\\[2mm]
> > > &\hspace{-130mm}\leq\left( 1 - \frac{\mu\tau_k}{2}\right) \lVert x_{k} - x^\*\rVert^2 +  \frac{1}{\beta}\lVert y_k - y^\* \rVert^2 + \tau_k L_k\lVert x_k - x_{k-1}\rVert^2 +2\tau_k\theta_k P_{x^\*, y^\*}(x_{k-1}).
> > > \end{align}
> > >
> > >
> > > Using e) and f) from the list we lower bound the coefficient of the second term in the main inequality above:
> > > \begin{align}
> > >      \hspace{30mm}\frac{\sigma_{\mathrm{min}}^2(A)\tau_k^3L_k}{1+2\tau_k^2L_k L} &\geq  \frac{\frac{\sigma_{\mathrm{min}}^2(A)}{4 \left(4L^2 + \beta\lVert A \rVert^2\right)} \frac{\mu}{2\sqrt{4\mu^2 + \beta\lVert A \rVert^2}}}{1  + \frac{L^2}{2\sqrt{4\mu^2 + \beta\lVert A \rVert^2}\sqrt{4L^2 + \beta\lVert A \rVert^2}}} \\\\[2mm]
> > >     &= \frac{\sigma_{\mathrm{min}}^2(A) \mu }{8\left( 4L^2 + \beta\lVert A \rVert^2\right) \sqrt{4\mu^2 + \beta\lVert A \rVert^2} + 4L^2\sqrt{4L^2 + \beta\lVert A \rVert^2}}\\\\[2mm]
> > >     &\textcolor{red}{=: T}
> > > \end{align}
> > >
> > >
> > > Using properties a), b), c) and d) in the list and ignoring the positive terms on the LHS without a correspondent on the RHS, the main inequality becomes:
> > > \begin{align}
> > > \lVert x_{k+1} - x^\*\rVert^2 + \left(\frac{1}{\beta} + T \right)\lVert y_{k+1}- y^\* \rVert^2 + \frac{1}{2}
> > > \lVert x_{k+1} - x_k \rVert^2& + 2\tau_k(1+\theta_k)  P_{x^\*, y^\*}(x_{k}) \\\\[2mm]
> > > &\hspace{-90mm}\leq\left( 1 - \frac{\mu\tau_k}{2}\right) \lVert x_{k} - x^\*\rVert^2 +  \frac{1}{\beta}\lVert y_k - y^\* \rVert^2 + \frac{1}{2}\left(1 - \frac{1}{2} \right) \lVert x_k - x_{k-1}\rVert^2 +2\tau_{k-1}(1 + \theta_{k-1}/2) P_{x^\*, y^\*}(x_{k-1}).
> > > \end{align}
> > >
> > >
> > > To simplify notations for writing the contractions for this inequality, denote $s = \sqrt{4L^2 + \beta\lVert A \rVert^2}$ and $t = \sqrt{4\mu^2 + \beta\lVert A \rVert^2}$. We thus have the following contractions:
> > >
> > >   * for $\frac{1}{2}
> > > \lVert x_{k+1} - x_k \rVert^2$ it is: $1 - \frac{1}{2}$;
> > >   * for $\left(\frac{1}{\beta} + T \right)\lVert y_{k+1}- y^* \rVert^2$ it is:
> > >    \begin{align}
> > >     \hspace{30mm}\frac{1}{1 + T\beta} &= 1 - \frac{T\beta}{1 + T\beta}\\\\
> > >     &= 1 - \frac{\beta \sigma_{\mathrm{min}}^2(A) \mu}{\beta \sigma_{\mathrm{min}}^2(A) \mu + 8s^2t + 4L^2s};
> > >   \end{align}
> > >   * for $\lVert x_{k+1} - x^*\rVert^2$ it is:
> > >   \begin{equation}
> > >     \hspace{30mm}1 - \frac{\mu\tau_k}{2} \leq 1 - \frac{\mu}{4s};
> > > \end{equation}
> > >   * finally, for $2\tau_k(1+\theta_k)  P_{x^\*, y^\*}(x_{k})$ it is:
> > >     \begin{align}
> > >  \hspace{30mm}\frac{1+\theta_{k-1}/2}{1 + \theta_{k-1}} &= 1 - \frac{\theta_{k-1}}{2(1 + \theta_{k-1})}   \\\\
> > >  &\leq 1 - \frac{t}{s} \tag{\small By def. of $\theta_{k-1}$ and property 5).\normalsize}.
> > > \end{align}
> > >
> > >
> > > Note that for  the latter two contractions above, it always holds that $\mu/ (4s) < t/s$ so in the final  bound  we can ignore the latter. Finally, denoting  the  LHS of the main inequality as $E_k$, we have that:
> > > \begin{equation}
> > >      \hspace{30mm}E_k \leq \left( 1 - \min \left\\{ \frac{1}{2}, \frac{\mu}{4s},\frac{\beta \sigma_{\mathrm{min}}^2(A) \mu}{\beta \sigma_{\mathrm{min}}^2(A) \mu + 8s^2t + 4L^2s} \right\\}\right)^k E_0
> > > \end{equation}
> > >
> > >
> > >
> > > These constants  now are of the right order and they scale with our min-max objective: $f(x) +\langle Ax, y\rangle - g^*(y)$. Specifically, $s$ and $t$ scale linearly. Note that if we only scale the corresponding primal objective $f(x) + g(Ax)$, then $A$ will not be scaled. However, we can use $\beta$ which always appears alongside $\lVert A \rVert^2$. So if we tuned $\beta$ for a given primal problem, which we  then scale by a factor $D$, we need  to simply do $\beta_{\mathrm{new}} = D^2\beta$.
> > >
> > >
> > > Finally, as a sanity check, if we set $A = 0$ the second constant in the rate becomes the same as the contraction factor in [1].
> > >
> > > We thank the reviewer again for helping us improve our paper.
> > >
> > > [1] Y. Malitsky and K. Mishchenko. Adaptive gradient descent without descent

---

> > > > ### Comment · Reviewer_prmj · 2021-08-23
> > > > **Thank you for improving!**
> > > >
> > > > Thank you for addressing my questions. I have tried to look into your proofs and believe something you are doing in the new analysis makes more sense as well. This has mostly addressed my concerns about the paper. Based on this I have raised my score for the paper.

---

> > > > ### Comment · Reviewer_prmj · 2021-08-25
> > > > **Further Follow-up Questions**
> > > >
> > > > Dear authors,
> > > >
> > > > I realized you haven't fully answered my first question regarding the discrepancy of the condition number you achieve using the adaptive method for this strongly-convex case (even using your new proof and result), compared with the non-adaptive prior work you mentioned in your first comment. Would be good if you provide some explanation why this is the case - i.e. is there a technical barrier for matching the rate $\max(\frac{L}{\mu}, \frac{\sigma_\mathrm{max}^2(A)}{\sigma_\mathrm{min}^2(A)})$ in the adaptive setting?
> > > >
> > > > Also, we found these two papers that look like closely related to you papers. Would you mind comparing your paper with these (in terms of results, techniques, and generality) in more detail?
> > > >
> > > > * Malitsky, Yura. "Golden ratio algorithms for variational inequalities." Mathematical Programming 184.1 (2020): 383-410.
> > > >
> > > > (more specifically, it looks to me like they also have an 1/T rate with similar constant as your work; it is unclear to me how their linear rate compares with yours.)
> > > >
> > > > * Ene, Alina, Huy L. Nguyen, and Adrian Vladu. "Adaptive Gradient Methods for Constrained Convex Optimization and Variational Inequalities." arXiv preprint arXiv:2007.08840 (2020).

---

> > > > > ### Author Response · Authors · 2021-08-28
> > > > > **Thank you for the follow-up**
> > > > >
> > > > > **Regarding the condition  number discrepancy.**
> > > > >     The reason why we are unable to achieve the same nice separation of condition numbers is two-fold (1) our iteration is set up in the style of CVA, and (2) we have essentially one stepsize to compute ($\tau_k$, since $\sigma_k = \beta \tau_k$), which thus will bear the burden of obeying the problem structure w.r.t. both $L$ and $\lVert A \rVert$.
> > > > >
> > > > >
> > > > > To illustrate the above, consider an alternative formulation of the stability condition for fixed stepsize CVA, by letting $\beta := \sigma/\tau$ where $\sigma$, $\tau$ are the fixed stepsizes. The stability condition can then be re-written as $\left( \frac{1}{\tau} - L\right)\frac{1}{\beta\tau} \geq \sigma_{\mathrm{max}}(A)^2$. Solving for $\tau$ we get that $\tau \in \left(0,  \frac{2}{L^2 + \sqrt{L + 4\beta\sigma^2_{\mathrm{max}}(A)}}\right)$, which is highly reminiscent of our local conditions for $\tau_k$. It seems thus that these stepsize conditions are tightly linked to the fact that we are extending CVA, since we arrive at a local version of this condition in our analysis despite taking a different route than e.g., [5].
> > > > >
> > > > >
> > > > > Keeping the above in mind, even when the contraction stems uniquely from strong convexity of $f$ for the term $\lVert x_{k+1} - x^\* \rVert^2$ (first equation below line 545 in the appendix), our constant will be $\frac{\mu \tau_k}{2} \geq \frac{\mu}{2 \sqrt{4L^2 + \beta \sigma^2_{\mathrm{max}}(A)}}$ as opposed to just  being proportional to $\mu/L$. Regarding obtaining a contraction for $\lVert y_k - y^\*\rVert^2$ (the more problematic term), the only place where we can do it given our energy function is from eq. (44)-(47) in the appendix. Since we have a factor of $\tau_k/L$ in front of the term $\lVert A(y_{k+1} - y^\*) \rVert^2 $ on the RHS, we cannot avoid the "mixed" contraction term, whichever setting of $\delta$ we choose. It seems that the analysis required to achieve this style of adaptivity comes at the cost of worse constants.
> > > > >
> > > > >
> > > > > PDFP$^2$O achieves this nice separation through a combination of a different iteration style than CVA, as well as performing a fundamentally different kind of analysis (they express their iteration in fixed-point form to show convergence). In this context they are also able to separate the stability conditions on the stepsizes -- specifically, $0 < \lambda \leq 1/ \sigma^2_{\mathrm{max}}(A)$ and $0 < \gamma < 2L$. A significant drawback of their type of analysis, however, is that their algorithm has no rate guarantees when $f$ is only smooth and not strongly convex (i.e. they only derived asymptotic convergence). Also, they require 3 matrix-vector multiplications per iteration whereas we only require 2.
> > > > >
> > > > >
> > > > > **Comparison with  [1].** We thank the reviewers for pointing this out. Indeed, this paper is related to ours in the way adaptivity is achieved, and addresses the more general setting of VI. Our problem formulation can be recovered from theirs by setting $u = (x, y)$,  $g(u) = g^*(y)$ and
> > > > >     \begin{align}
> > > > >         F(u) = F((x, y)) = \begin{bmatrix}
> > > > >            \nabla f(x) + A^Ty \\\\
> > > > >            -A^Tx
> > > > >          \end{bmatrix}.
> > > > >     \end{align}
> > > > >
> > > > >
> > > > > Note that  for $\nabla f$ Lipschitz continuous with constant $L_f$ and strongly convex with constant $\mu$, $F$ is Lipschitz continuous with constant $L_{F} = \sqrt{2\left( L_f^2 + \sigma^2_{\mathrm{max}}(A)\right)}$ and strongly monotone with (the same) constant $\mu$, respectively.
> > > > >
> > > > > The  algorithm proposed in [1] sets the stepsize $\lambda_k$ depending on the local Lipschitz constant of $F$, in a similar fashion to us and [4]. The differences are:
> > > > >   * $\lambda_k$ can be at most $\rho$ times bigger than $\lambda_{k-1}$, where $\rho$ is a fixed constant given as input. In our case, $\rho = \rho_k = \sqrt{1 + \theta_k}$ -- so the increase is itself adaptive;
> > > > >   *  an upper bound $\bar{\lambda}$ on the stepsize $\lambda_k$  is required as input to the algorithm and is said to be chosen "quite large" (bottom of page 5). Ideally, $\bar{\lambda}$ is proportional to the upper-bound on the inverse local Lipschitz constant over the iterate sequence, but this information cannot be known in advance.
> > > > >
> > > > > For the setting of $F$ locally Lipschitz continuous, the authors show ergodic convergence in eq. (39) with a rate of
> > > > >     \begin{equation*}
> > > > >        \hspace{30mm}\frac{M}{\sum_{i = 1}^k\lambda_i} \leq  \frac{(16 /9) M L_F^2 \bar{\lambda} }{k} = \frac{(32 /9) M \left( L_f^2 + \sigma^2_{\mathrm{max}}(A)\right) \bar{\lambda} }{k},
> > > > >     \end{equation*}
> > > > > where the inequality comes from $\lambda_k \geq \frac{\phi^2}{4L_F^2\bar{\lambda}}$ in eq. (25) and they suggest $\phi = 3/2$. In the bound, $M$ contains initial distances to the optimum point. The bound has worse constant dependence than ours in terms of $L_f$ and $\sigma_{\mathrm{max}}(A)$, and it depends on $\bar{\lambda}$ for which there is  no guidance on how to set -- we suppose it could be treated as a hyper parameter, though it is not specified in the experimental section.
> > > > >
> > > > > The linear convergence scenario is proven under an error bound assumption (see eq. (42) and the following paragraph), which should hold for all iterates after a certain point $k_0$ in the optimization process. The contraction constant $m$ is not explicitly given, but merely shown to exist (see the first two lines below eq. (44) on page 9).
> > > > >
> > > > > We can, however, try to particularize this proof to our problem setting. First, let us rename the constant $\mu$ in inequality (42) of the error bound to $\gamma$, so as to avoid confusion with our strong convexity/monotonicity constant. When $F$ is strongly monotone and Lipschitz continuous, Theorem 3.1 in [3] shows that the error bound holds for any point with $\gamma = \frac{L_F + 1}{\mu}$. Thus, $k_0$ in the proof of [1] can be set to 0. What is left is to approximate the order of $m$ and we use the following estimation:
> > > > >  * We multiply inequality (44) with $m\beta\gamma^2$ and ask that the RHS of the resulting inequality be smaller than the  RHS of (45).
> > > > >  * Identifying term-for-term, we thus ask that $2m\beta\gamma^2 (1 + \lambda_kL_F)^2 \leq \theta_k$. Using the definitions of $\beta$, $\theta_k$, $\gamma$, and setting $\phi = 3/2$ as suggested, then upper bounding the LHS and lower bounding the RHS based on the upper and lower bounds on $\lambda_k$ in Lemma 2 of [1], we arrive at:
> > > > > \begin{align*}
> > > > >  &\hspace{0mm} \frac{4m(1 + L_F)^2(1 + \bar{\lambda}L_F)^2}{\mu^2} \leq \frac{9}{16 L_F^2\bar{\lambda}^2} \iff    \\\\
> > > > >  &\hspace{0mm}m \leq \frac{9\mu^2}{64(1+L_F)^2 (1+\bar{\lambda}L_F)^2L_F^2\bar{\lambda}^2} \overset{\mathrm{def.} L_F}{\iff}\\\\
> > > > >  &\hspace{0mm} m \leq \underbrace{\frac{9\mu^2}{128\left(1+\sqrt{2\left( L_f^2 + \sigma^2_{\mathrm{max}}(A)\right)}\right)^2 \left(1+\bar{\lambda}\sqrt{2\left( L_f^2 + \sigma^2_{\mathrm{max}}(A)\right)}\right)^2\left( L_f^2 + \sigma^2_{\mathrm{max}}(A)\right)\bar{\lambda}^2}}_{\textcolor{red}{=:T}}
> > > > > \end{align*}
> > > > >
> > > > > With this approximation, the contraction will be at best $(1 - T)$ (since $m \leq T$), which is a worse constant than ours. The author of  [1] nevertheless states that `we are not interested in sharp constants, but rather in showing the linear convergence' -- so it's likely that these numbers do not reflect the practical performance of the algorithm, which seems to be good.
> > > > >
> > > > > In conclusion, [1] presents an algorithm for generic VI problems which uses local operator information to estimate the inverse Lipschitz constant for the adaptive stepsizes. A nice property of this algorithm is that it does not require $\lVert A \rVert$ and has good practical performance in the experiments. However, since the VI framework is very general and does not take full advantage of the problem structure (e.g. the fact that $\langle Ax,y\rangle$ is a bilinear term), it naturally comes with worse bounds than algorithms specifically designed to solve our template.
> > > > >
> > > > > It was an omission on our side to not include this work as a precursor to both [4] and our work, and we will rectify this.
> > > > >
> > > > > **Comparison with [2].** This work does not seem to bear a close relation to ours beyond the point of an automatic setting of the stepsizes (different in nature from APDA). Indeed they consider a VI framework that can cover convex-concave problems, but their problem formulation is constrained to a set $\mathcal{K}$.
> > > > >
> > > > > Importantly, the algorithm given in Figure 4, requires the knowledge of $R_\infty \geq \max_{u, v \in K} \lVert u - v\rVert_\infty$ for setting the stepsize; thus the adaptivity is somewhat partial, as it depends on global information. Additionally, the requirement of $R_\infty$ implies a requirement for bounded domain (both primal and dual, in our case), which is more restrictive than our setting; for APDA we prove the sequence boundedness without any assumption on the domain. Finally, the rate given for Lipschitz continuous $F$ (set $\beta_i = 1$ to match our setting) in Thm 3.7 is:
> > > > >      \begin{equation*}
> > > > >          \hspace{30mm}\mathcal{O}\left(\frac{R_{\infty}^2d\ln(2) }{k} \right),
> > > > >      \end{equation*}
> > > > >      where $d$ is the problem dimension, which is not directly comparable to ours in terms of constants.
> > > > >
> > > > > Under these arguments the methods are, in  our opinion, not significantly related.
> > > > >
> > > > > [1] Y. Malitsky. "Golden ratio algorithms for variational inequalities." (2020)
> > > > >
> > > > > [2] A. Ene et al. "Adaptive Gradient Methods for Constrained Convex Optimization and Variational Inequalities."
> > > > >
> > > > > [3] J. Pang. "A Posteriori Error Bounds for the Linearly-Constrained Variational Inequality Problem". (1987)
> > > > >
> > > > > [4] Y. Malitsky and K. Mishchenko. Adaptive gradient descent without descent
> > > > >
> > > > > [5] A. Chambolle, T. Pock. "On the ergodic convergence rates of a first-order primal–dual algorithm", (2016)

---

### Official Review · Reviewer_EuGK · 2021-07-18

**Rating:** 7
**Confidence:** 4

**Summary:**

This paper studies adaptivity of first-order primal-dual methods with respect to the smoothness or local smoothness of the problem. The manuscript builds upon a previous publication (Yura Malitsky and Konstantin Mishchenko. Adaptive gradient descent without descent. In Interna347 tional Conference on Machine Learning, pages 6702–6712. PMLR, 2020) to primal dual setup. Ergodic convergence is shown for the proposed method, where no knowledge of L is needed. The results is based on local smoothness which is defined for pairs of iterates of the sequence of points generated by the algorithm. The case of strongly convex function is also studied.

**Main Review:**

This is a good paper, it is easy to follow and many of the concepts and ideas and nicely presented. Also, the results are interesting and will provide a good contribution to adaptive methods. I did not checked every line of the proofs but spend a couple of hours going over them and I believe they are correct.

I have a couple of comments that would like to be addressed.

Abstract: some quantification of "faster convergence" would be useful, what minor change?

Some more formality is needed in the definition of locally smoothness. Line 121 page 3, do the authors mean there is a finite L such that for each compact subset the gradient is Lipschitz? or for every compact subset one can find a finite L. This might need clarification, because then the example provided is smoothness with respect to two iterates, which is a very local and path dependent constant. An identical comment follows for the local strong convexity property. I believe these items are of crucial importance in the analysis so maybe a formal definition would be great.

Some discussion about the fact the the final bounds are given with respect to the smoothness, (and strong convexity) of the function on the convex hull of the iterates. This is on, but some discussion is needed. For example, does such L exists for any initial point? At the end of the day, we such convergence bounds to estimate the number of iterations required to get some \epsilon accuracy, and then we will inevitably need some estimate of L. This L in theory depends on all the iterates, all the way to infinite iterates, is there a way to estimate this?

Maybe some extra discussions is needed about the difference between having a function being smooth, and it being locally smooth for all possible sequences of the algorithm for different initial points. I believe this is the critical point that will show the weaker assumptions required in this manuscript.

minor typo in like 119 page 3 "it its gradient"

minor: it looks a bit weird to have the algorithm in between a section and a subsection floating around.

**Time Spent Reviewing:**

4

---

> ### Author Response · Authors · 2021-08-10
> **Thank you for the feedback and suggestions**
>
> We thank the reviewer for their time, their thoughtful comments and the helpful suggestions for improving our paper. We provide clarifications point by point below.
>
> * **Regarding the abstract:** By “faster convergence” we meant that, under the additional assumption of locally strongly convex $f$ and full row rank $A$, the convergence rate of APDA can be improved (i.e. become linear) compared to the base case in which APDA achieves only sublinear convergence. By “minor change” we referred to the change in stepsize between Thm 4.1 and Thm 4.2, namely that the first term under the square root of Eq. (5) becomes $(L_k + 1/L_k)^2$ for Thm. 4.2 instead of just $L_k^2$. We will rephrase for clarity.
>
>
> * **Regarding the lack of formality for defining local Lipschitzness:** Indeed, we were remiss to not include the formal definitions, which we list below.
>
>     **Def.1.** A function $f: \mathbb{R}^d \to \mathbb{R}$, is locally Lipschitz-smooth if for any compact subset $\mathcal{C} \subset \mathbb{R}^d$ there exists $L_{\mathcal{C}} > 0$ such that
>     $$
>     \lVert \nabla f(x) - \nabla f(y)\rVert \leq L_{\mathcal{C}} \lVert  x - y \rVert,   \forall x, y \in \mathcal{C}.
>     $$
>
>     **Def.2.** A function $f: \mathbb{R}^d \to \mathbb{R}$, is locally strongly convex if for any compact subset $\mathcal{C} \subset \mathbb{R}^d$ there exists $\mu_{\mathcal{C}} > 0$ such that
>     $$
>     f(x) \geq f(y) + \langle\nabla f(y), x - y\rangle + \frac{\mu_{\mathcal{C}}}{2} \lVert x-y \rVert^2,  \forall x, y \in \mathcal{C}.
>     $$
>
> * **Regarding the Lipschitz constant in the final bounds:** The $L$ constant present in our final bounds exists for any initial points $x_0$, $y_0$. This follows from the proven boundedness of the iterates (Thm 4.1 point 1) which is independent of the initial points. The existence of $L$ is then a direct consequence of this fact and the local Lipschitzness of $\nabla f$.
>
>   Regarding the estimation of $L$ as defined in our paper, it is unfortunately generally not possible, apart from the trivial case where $f$ is $L$-smooth. One can possibly devise some heuristics for this purpose, however, to our knowledge there are no methods to guarantee the quality of approximation. This problem, however, would be faced by any method optimizing $f$ with a locally Lipschitz continuous $\nabla f$, as the value of $L$ is dependent on the trajectory towards the optimum. However, even having an accurate estimate of $L$ would be of only marginal help since the bound depends on $ \lVert x_1 - x^* \rVert$ and  $  \lVert  y_1 - y^*  \rVert$ quantities which are unknown by the very nature of what we are trying to achieve -- knowledge of $x^*$.
>
>   We thank  the reviewer for raising the question of estimating $L$, as this can be  a nice direction  for future research.
>
>
> * **Regarding $f$ being smooth versus locally smooth:** We can add a discussion after the definition of local smoothness and illustrate that our assumption is weaker on a simple example like $f(x) = \exp(x)$. We can also elaborate on the behaviour APDA in regions with large versus small $L_k$.
>
>    We hope to have understood correctly the kind of discussion the reviewer had in mind, but if  that's not the case we'd be happy to provide further (or different) clarifications.

---

### Decision · Program_Chairs · 2021-09-27

**Decision:**

Accept (Poster)

**Comment:**

The paper proposes an adaptive variant of the Condat-Vu algorithm for composite bilinear saddle-point problems, providing theoretical guarantees and reporting some promising experimental results. After fairly extensive discussion, the reviewers have converged to a positive consensus on the paper, and I consequently recommend its acceptance.

Based on the discussion, the authors should carefully revise their manuscript, paying close attention to the following points:

* Comparison with the related works mentioned by the reviewers.

* Clearing-up of the derivation of the strongly-convex case, aiming for a concise bound easily comparable with the best known non-adaptive bound.

* Inclusion of additional results with more extensive tuning of the non-adaptive baseline.